# Modeling human telencephalic development and autism-associated SHANK3 deficiency using organoids generated from single neural rosettes

Human telencephalon is an evolutionarily advanced brain structure associated with many uniquely human behaviors and disorders. However, cell lineages and molecular pathways implicated in human telencephalic development remain largely unknown. We produce human telencephalic organoids from stem cell-derived single neural rosettes and investigate telencephalic development under normal and pathological conditions. We show that single neural rosette-derived organoids contain pallial and subpallial neural progenitors, excitatory and inhibitory neurons, as well as macroglial and periendothelial cells, and exhibit predictable organization and cytoarchitecture. We comprehensively characterize the properties of neurons in SNR-derived organoids and identify transcriptional programs associated with the specification of excitatory and inhibitory neural lineages from a common pool of NPs early in telencephalic development. We also demonstrate that neurons in organoids with a hemizygous deletion of an autism- and intellectual disability-associated gene *SHANK3* exhibit intrinsic and excitatory synaptic deficits and impaired expression of several clustered protocadherins. Collectively, this study validates SNR-derived organoids as a reliable model for studying human telencephalic cortico-striatal development and identifies intrinsic, synaptic, and clustered protocadherin expression deficits in human telencephalic tissue with *SHANK3* hemizygosity.

The telencephalon is the largest and the most evolutionary advanced region in the human brain[1–3]. Early in development, it is divided into the dorsal and ventral telencephalon (pallium and subpallium, respectively) that give rise to several interconnected anatomical structures, including the cerebral cortex and striatum, that have been associated with many human-specific behaviors and disorders[4,5]. The composition of cells in the developing human cerebral cortex and striatum have been extensively investigated in recent studies using single cell RNA sequencing, providing insights into the increased diversity and unique molecular properties of human telencephalic

cells[6–10]. However, our understanding of the molecular mechanisms associated with the specification of different types of human telencephalic cells under normal conditions and in patients with genetic neurodevelopmental disorders remains very limited[11,12].

Brain region-specific organoids produced from human pluripotent stem cells offer an enticing approach to model the cellular and molecular aspects of human brain development in vitro[13–22]. Indeed, it has been demonstrated that human-specific aspects of telencephalic neurogenesis, migration, and gliogenesis are largely recapitulated in organoids and that organoids have similar transcriptional and

e-mail: alexsh@neuro.utah.edu

epigenetic profiles to primary fetal human neocortical tissue. In addition, it has been shown that different region-specific organoids can be fused together or transplanted to the mouse brain for studying inter-regional neuronal migration and anatomical and functional neuronal connectivity[23–30]. Importantly, organoids have also been produced from patient-derived and engineered stem cell lines and used to study the cellular and molecular deficits in human neurodevelopmental disorders associated with autism and malformations of cortical development[15,17,26,31–33], highlighting the potential of brain organoids to revolutionize patient-specific drug discovery[34].

Despite substantial progress that has been made using stem cell-derived brain organoids, there are still limitations, including an unpredictable number and organization of germinal zones within an organoid, impaired specification of certain important telencephalic cell types[35], and unknown effects of the chemicals used in some differentiation protocols, such as Wnt inhibitors for an efficient specification of cortical cells in iPSC-derived organoids[18,20,36] or retinoic acid (RA) receptor agonists for the specification of striatal cells in organoids[30], on normal development and disease-associated phenotypes.

In the present study, we generated telencephalic organoids from isolated self-organized single neural rosettes (SNRs) without using any potent agonists/antagonists of brain morphogenesis. It has previously been demonstrated that the organization and properties of cells in rosettes closely resemble those in the anterior neural tube[37–40]. We demonstrated that SNR-derived organoids maintain the SNR-based self-organization and reproducibly generate different types of predictably organized telencephalic cells, including pallial and subpallial NPs, cortical excitatory neurons (EN) and LGE-derived inhibitory neurons (IN), astrocytes, oligodendrocytes, and ependymal and peri-endothelial cells. In addition, we demonstrate that SNR-derived organoids contain functionally mature neurons and neural networks capable of generating diverse electrical signals and oscillatory rhythms. Finally, we used SNR-derived organoids to investigate the cellular and molecular deficits caused by the deletions of the autism- and intellectual disability-associated gene *SHANK3*[41,42]. Together, our results suggest that SNR-derived organoids can be used as a reliable platform for studying human telencephalic development and the cellular and molecular deficits in human genetic neurodevelopmental disorders. An interactive visualization of the transcriptomic and electrophysiological data obtained in this study is provided in an online browser (UBrain Browser: http://organoid.chpc.utah.edu).

## Results

### Generation of organoids from single neural rosettes

We isolated SNRs from the clusters of neural rosettes that were obtained from multiple human pluripotent stem cell (PSC) lines (Supplementary Table 1) using a modified dual-SMAD inhibition protocol for neural induction[43] and post-induction treatment with epidermal growth factor (EGF) and basic fibroblast growth factor (bFGF) for the amplification of NPs (Supplementary Fig. 1). SNRs demonstrated consistent size and organization (Supplementary Fig. 2) and were characterized by expression of the telencephalic marker FOXG1, NP markers PAX6 and SOX2, and proliferation markers KI67 and PH3 (Supplementary Fig. 3). SNRs also contained few cells expressing a neuronal marker MAP2 (Supplementary Fig. 3) and almost no cells expressing a subpallial medial ganglionic eminence (MGE) marker NKX2.1 (Supplementary Figs. 3, 4). These results indicated that PSC-derived SNRs predominantly consist of pallial or pallial-subpallial telencephalic NPs.

SNR-derived organoids generated from six human PSC lines demonstrated consistent growth and organization across multiple differentiation batches (Fig. 1a–c and Supplementary Fig. 5). A clearly visible single lumen, resembling a neural tube- or ventricle-like structure, was detected in ~50% of SNR-derived organoids (Supplementary

Fig. 5e). Immunostaining with cell-type specific antibodies showed that most cells in SNR-derived organoids expressed the telencephalic marker FOXG1 (Fig. 1d and Supplementary Fig. 6b), with pre-dominantly nuclear expression patterns observed in NPs and both nuclear and cytoplasmic expression observed in neurons[44]. The cells immediately adjacent to the lumen expressed N-cadherin (Supplementary Fig. 6a), NP markers SOX2 and PAX6 (Fig. 1e, f), proliferation markers KI67 (Fig. 1d) and PH3 (Fig. 1h), and a radial glial (RG) marker phospho-vimentin (pVIM) (Fig. 1h, i), while the cells distributed more peripherally to the lumen expressed the marker of Cajal-Retzius cells, Reelin, and markers of both ENs and INs, including MAP2, CTIP2, TBR1, and GAD67 (Fig. 1e–g, and Supplementary Fig. 6c, d). We also found a relatively small proportion of cells expressed a marker of intermediate progenitors (IPs) TBR2 (Supplementary Fig. 7b) and no cells expressed the markers of ventral MGE NPs NKX2.1 (Supplementary Fig. 6b) or posterior NPs, such as EN1, FOX2A, or PAX3 (Supplementary Fig. 8). Interestingly, mitotic RG, unipolar cells co-expressing both PH3 and pVIM (~0.5% of cells, Fig. 1j), were found on both the apical and basal sides of the ventricular-like zone surrounding the lumen (Fig. 1h). However, both the apical and basal RG showed detectable expression of HOPX (Fig. 1i), indicating an early developmental status of 1-month-old SNR-derived organoids[6,45,46]. Together, these results demonstrate that telencephalic organoids with predictable organization of NPs and neurons around the lumen can be reliably generated from isolated SNRs.

To further characterize the identities of cells in 1-month-old SNR-derived organoids and the consistency of organoid cellular composition, we performed single-cell RNA-sequencing (scRNA-seq) on 9577 cells collected from four 1-month-old SNR-derived organoids produced from two different PSC lines in separate differentiation batches (Fig. 2a). Using unsupervised clustering, we identified 15 molecularly distinct cell clusters (Fig. 2b, c) that were characterized by the expression of specific marker genes (Supplementary Fig. 9 and Supplementary Data 1). We also found that most cell clusters were composed of cells originating from multiple organoids (Fig. 2d, e), suggesting the reproducibility of cellular composition across organoids and protocol consistency.

We confirmed the telencephalic identities of SNR-derived organoids, as most cells exhibited *FOXG1* expression (Fig. 2f, g) and no or undetectable expression of typical endodermal-, mesodermal-, or posterior brain-specific markers (Supplementary Fig. 10). Based on the gene expression profiles, all cells in SNR-derived organoids could be divided into five groups (Fig. 2b, c): NPs, characterized by the expression of NP markers *NES, SOX2, MKI67, PAX6, EMX2*, and *LHX2* (NP1-5 clusters); excitatory IPs (IP-EN), characterized by *NEUROG1* and *EOMES* expression; inhibitory IPs (IP-IN), characterized by *ASCL1, DLX1*, and *DLX2* expression; and newborn ENs (nEN) and INs (nIN), characterized by *SLC17A6* and *TBR1* and *GAD1, GAD2*, and *SLC32A1* (or VGAT) expression, respectively (Fig. 2f, g). Interestingly, we found that cells in the NP5 cluster showed increased expression of the typical outer RG (oRG) markers, including *SLC1A3, HOPX, PTPRZ1, LIFR*, and *MOXD1*[6,45,47] and that cells in the NP4 cluster were characterized by an elevated expression of typical LGE markers, such as *GSX2, SOX6, DLX1*, and *DLX2*, but not MGE or caudal ganglionic eminence (CGE) markers *NKX2.1* and *LHX6* or *NR2F2* and *HTR3A*, respectively. These results suggest that different subtypes of excitatory and inhibitory NPs and neurons with telencephalic identity are consistently generated in 1-month-old SNR-derived organoids.

#### Specification of excitatory and inhibitory lineages

The specification of both excitatory and inhibitory neural lineages in 1-month-old organoids provides an opportunity to investigate the transcriptional programs associated with the specification of these lineages in early human telencephalic development. We performed RNA velocity analysis, Velocyto[48], on scRNA sequencing data obtained

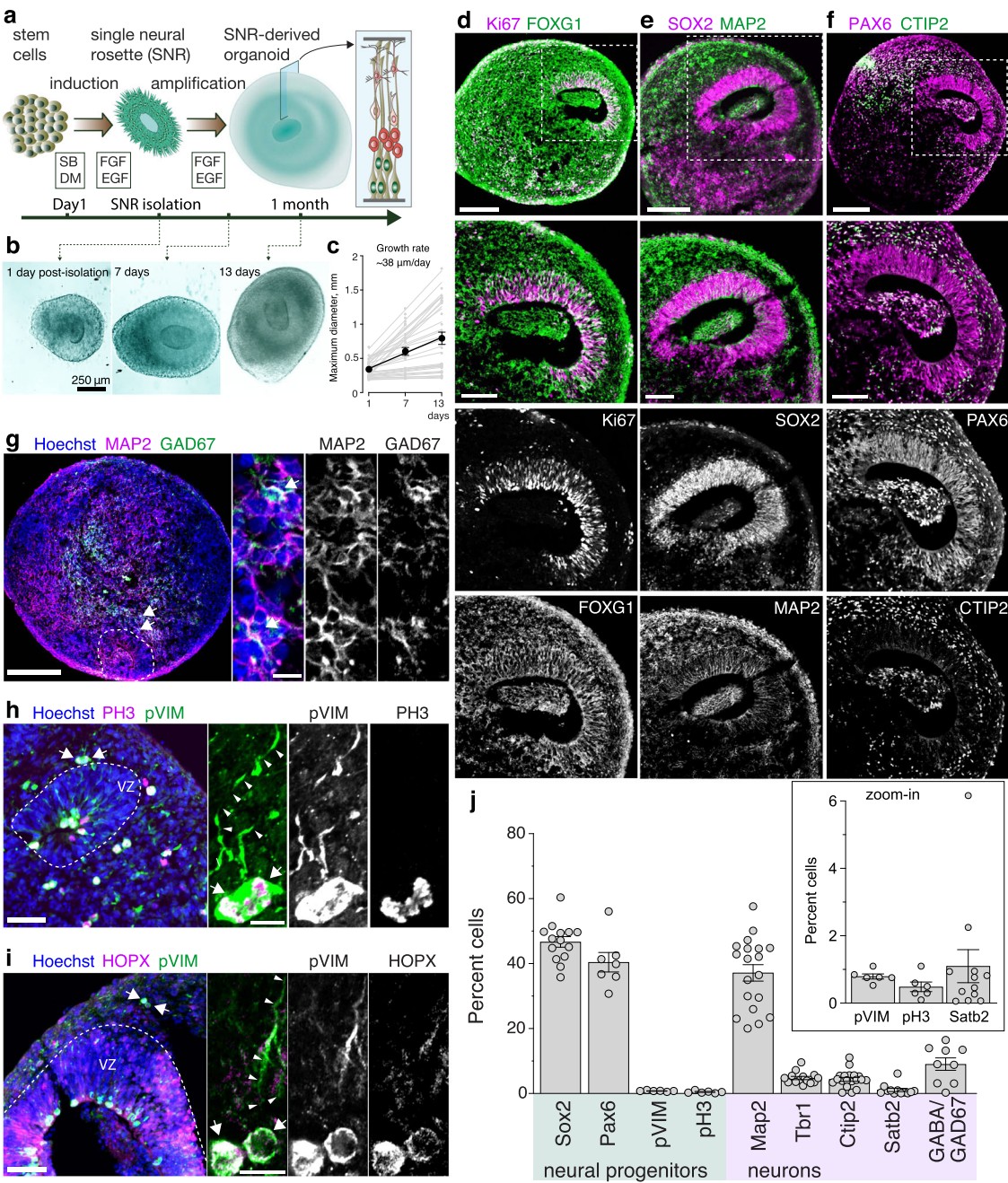

**Fig. 1 | Generation and characterization of 1-month-old SNR-derived organoids.**
**a** Protocol schematic (elements were drawn by Coni Hoerndli Science Design).
**b** Images of SNR-derived organoids at days 1, 7, and 13 post SNR isolation.
**c** Quantification of initial organoid growth (*n* = 32 rosettes obtained from H9 and 2242-5 lines). The line shows linear fit of the data. **d–i** Images of organoid sections immunostained for FOXG1 and Ki67 (**d**), SOX2 and MAP2 (**e**), PAX6 and CTIP2 (**f**), MAP2 and GAD67 (**g**), PH3 and pVIM (**h**) HOPX and pVIM (**i**). **j** Quantification of cells expressing different cell type-specific markers in organoid sections (*n* = 17 [SOX2], 7 [PAX6], 6[pVIM] 6[pH3], 19 [MAP2], 14 [TBR1], 19 [Ctip2], 12 [Satb2], 8 [GABA or GAD67] organoids/1–10 sections per organoid, produced in 1–4 differentiation batches from 2242-5, EP2-15, H9, and EYQ2-20 lines). Data are presented as mean ± standard error of the mean (s.e.m.). Scale bars = 250 (**b**), 200 and 50 (zoom-in) (**d–f**), 200 and 10 (zoom-in) (**g**) and 50 and 10 (zoom-in) (**h–i**) μm. Source data are provided as a Source Data file.

from 1-month-old organoids (Fig. 3a, Supplementary Fig. 11, and Supplementary Data 2). It predicts the dynamics and directions of development in a complex tissue based on the relative abundance of spliced and unspliced mRNA transcripts, and it has already been validated in multiple previous studies[8,48–50]. As a result of this analysis, NP1 cluster was identified as the most likely origin cluster in 1-month-old SNR-derived organoids (Fig. 3b). Interestingly, cells in this cluster were characterized by the expression of several Wnt pathway-associated genes, including *RSPO1*, *GPC3*, and *FZD5* (Fig. 3h and Supplementary Data 3), which is consistent with the idea that Wnt signaling plays an

important role in regulating the properties of NPs in human telencephalic development[51,52]. This analysis also identified the specific transcriptional changes associated with the specification and development of the EN and IN lineages in SNR-derived organoids (Fig. 3c). The specification of the IN lineage (NP1–NP4–IP–IN–nIN1-5) was associated with reduced expression of the Wnt pathway genes *RSPO1*, *GPC3*, and *FZD5* and increased expression of the Wnt antagonists *FRZB* and *SFRP1* and LGE-specific genes, including *GSX2*, *SOX6*, and *SALL3* (Fig. 3d, e and Supplementary Data 2). In contrast, the specification of the EN lineage (NP1–NP2/3–IP–EN–nEN1-3) was not associated with

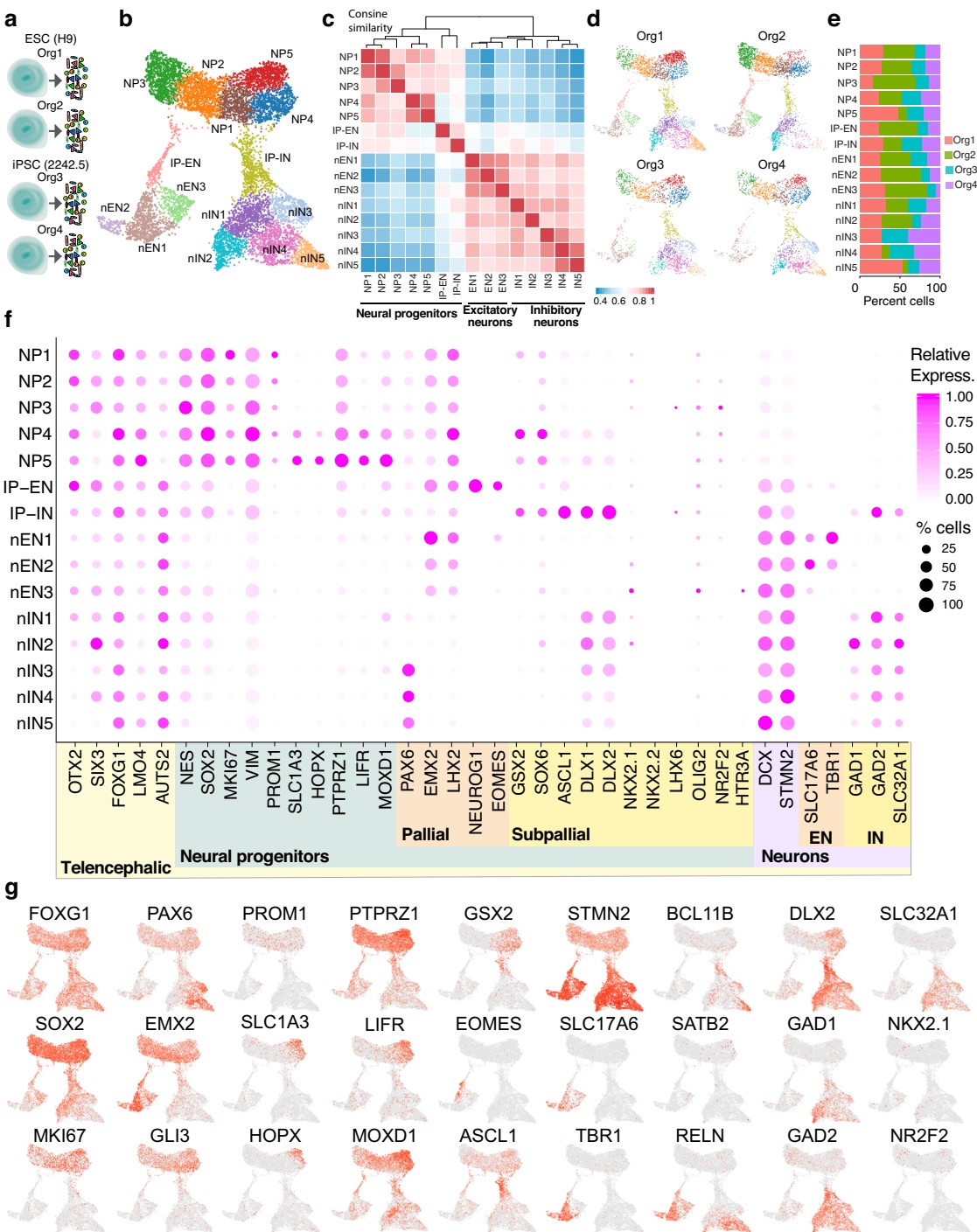

**Fig. 2 | 1-month-old SNR-derived organoids consist of telencephalic neural progenitors and both excitatory and inhibitory neurons. a** Cartoon depicting collection of single cells from individual organoids for single-cell RNA sequencing (elements were drawn by Coni Hoerndli Science Design). **b** UMAP visualization of unsupervised cell clustering (n = 4 organoids from H9, and 2242·5 lines). **c** Heatmap of relative gene expression similarity between different cell clusters. **d** UMAP visualization of unsupervised cell clustering from individual organoids. **e** Percentages of cells from different organoids in different cell clusters. **f** Dot plot visualization of expression of different cell type-specific markers. **g** Heatmap visualizations of expression of different cell type-specific markers. NP neural progenitor, IP intermediate neural progenitor, nEN newborn excitatory neuron, nIN newborn inhibitory neuron.

changes in the expression of the Wnt pathway genes (Fig. 3f–h and Supplementary Data 3), but it was associated with an elevated expression of the BMP pathway-associated genes, including *BMP7*, *BMPR1A*, *BMPR1B*, and *ID3* (Fig. 3i and Supplementary Data 3). These results support the idea that the Wnt and BMP signaling pathways play central roles in regulating the pallial-subpallial regionalization and the

specification of the EN and IN lineages in the developing human telencephalon[53] (Fig. 3j).

To gain further insights into the identities of the subpallial NPs and INs generated in 1-month-old organoids, we performed a comparative analysis of differentially expressed genes (DEG) in each organoid cluster with those that are differentially expressed in the fetal

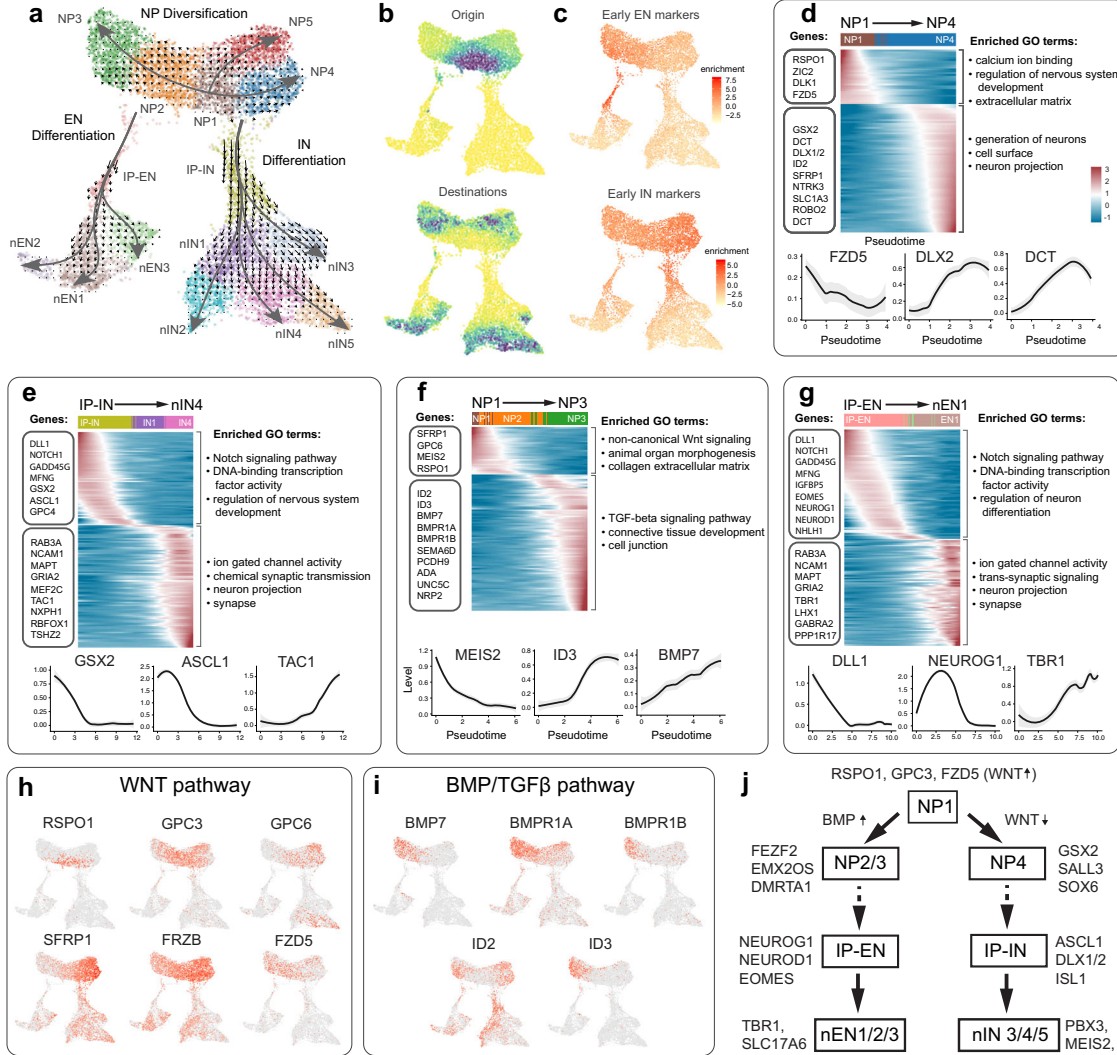

**Fig. 3 | Transcriptional programs and signaling pathways associated with specification of excitatory and inhibitory neural lineages in 1-month-old SNR-derived organoids. a** RNA velocity vector fields and developmental trajectories. Small arrows indicate extrapolated future states of cells. Large arrows indicate branching lineage trajectories constructed using Slingshot and initialized using the origin and end states of the Markov simulation. **b** The origin and end states of the differentiation landscape determined using Markov random walk simulation on the velocity field. The density of cells at the end of the simulation is shown using the color scale ranging from yellow (low) to blue (high). **c** Enrichment of early differentiation genes from the EN and IN differentiation trajectories. **d**–**g** Smoothed gene expression heatmaps of the top 150 genes differentially expressed along the NP1–NP4 (**d**), IP-IN–nIN4 (**e**), NP1–NP3 (**f**), and IP-EN–nEN1 (**g**) trajectories. Genes are ordered by peak expression time on the pseudotime axis. Selected genes are listed on the left and shown at the bottom. The top gene ontology terms are listed on the right. **h**–**i** UMAP visualization of expression of WNT (**h**) and BMP/TGFβ (**i**) signaling pathways genes expression. **j** Cartoon depicting selected genes and signaling pathways associated with specification of EN and IN lineages in SNR-derived organoids. NP neural progenitor, IP intermediate neural progenitor, nEN newborn excitatory neuron, nIN newborn inhibitory neuron.

human cortex (CTX), LGE, or MGE at 7, 9, and 11 PCW[8] (Fig. 4a and Supplementary Data 4). The cells in NP4 cluster showed the highest similarity with fetal LGE at 7PCW while cells in IN3-5 were most similar to fetal LGE at 9PCW, suggesting that these cells have LGE-like identities. Consistent with this idea, NP4 cells demonstrated increased expression of the LGE-specific NP markers *SALL3*, RP11-849I19.1 (LINC01896) and *GSX2*[8], and IN3-5 cells were characterized by an elevated expression of the markers of LGE-derived inhibitory striatal projection neurons (ISPN)[8], including markers *MEIS2*, *FOXP2*, and *PBX3* (Fig. 4b). We confirmed the presence of LGE-like NPs and LGE-derived ISPNs in SNR-derived organoids using immunostainings with anti-PAX6/GSX2 (Fig. 4c–e) and CTIP2/GABA antibodies (Fig. 4h, i), respectively. PAX6-expressing cells were evenly distributed throughout the lumen-surrounding VZ-like area, while GSX2- and PAX6/GSX2-expressing cells demonstrated a regionalized distribution (Fig. 4c–e). As we detected no GSX2-expressing cells in the pallium of the fetal

human telencephalon at PCW9 or 11 (Fig. 4f, g), INs in organoids were likely generated from subpallial LGE-like NPs. GABA/CTIP2-expressing cell in organoid sections were largely found inside of the organoids (Fig. 4h). Collectively, these results demonstrate that both pallial and subpallial LGE NPs are specified in 1-month-old SNR-derived organoids. The specifications of these NPs are associated with the changes in endogenous Wnt and BMP agonists/antagonists. These NPs give rise to both cortical ENs and LGE-derived INs in SNR-derived organoids.

## Reproducible generation of neuronal and glial cells

To encourage further development and maturation of neural cells in organoids, we propagated 1-month-old organoids embedded in Matrigel (MG) for an additional 4 months (Supplementary Fig. 1). MG-embedded organoids, produced from different PSC lines in different differentiation batches, demonstrated substantial expansion in size, reaching 3–4 mm in diameter by 5 months post induction (Fig. 5a–f).

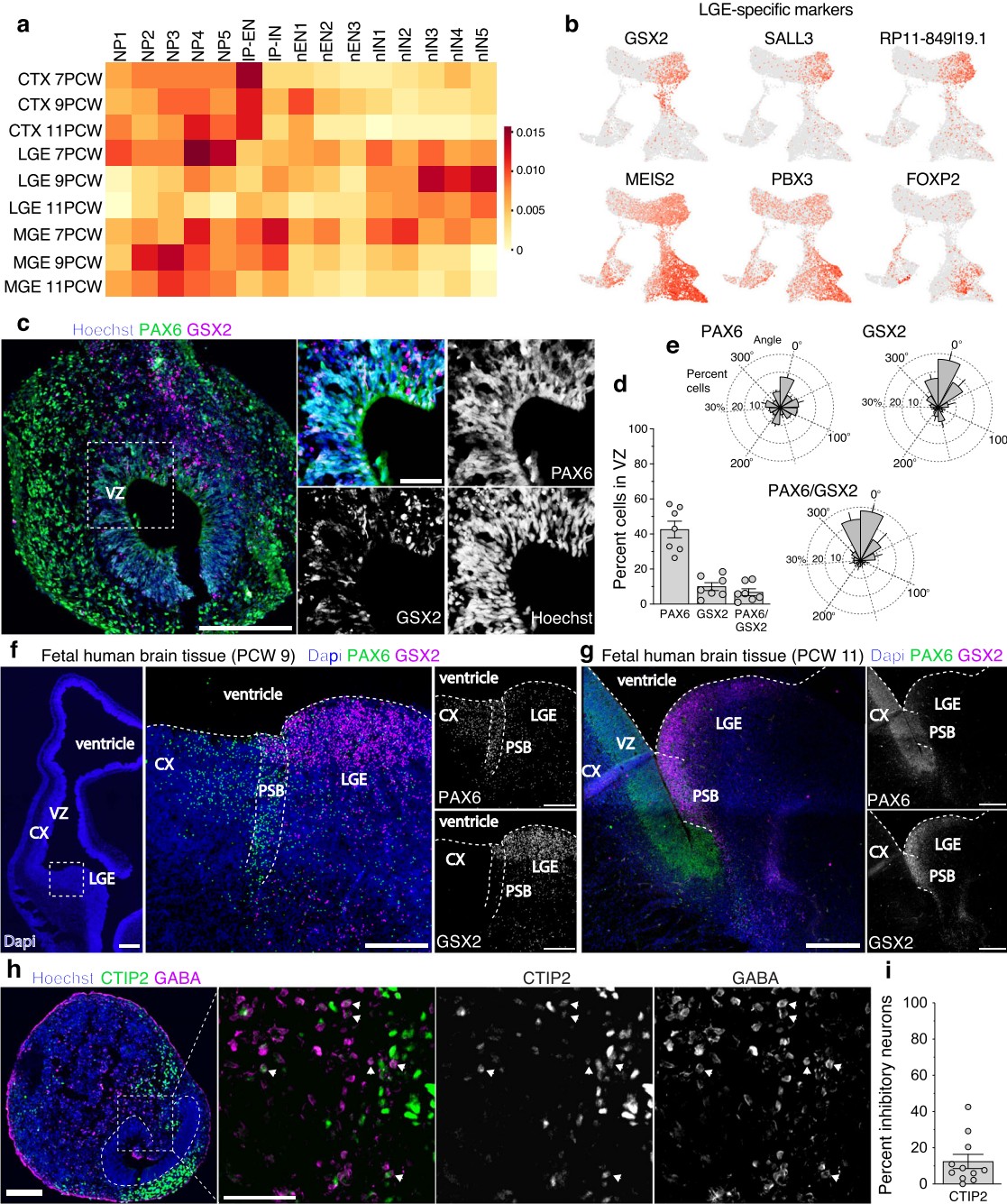

**Fig. 4 | LGE-like origin of inhibitory neurons in 1-month-old SNR-derived organoids. a** Heatmap visualization of the Jaccard similarity index between unique DEGs in the human fetal neocortex (CTX), lateral ganglionic eminence (LGE) and medial ganglionic eminence (MGE) at 7, 9, and 11 post-conception weeks (PCW)[8] and those in different clusters of 1-month-old SNR-derived organoids (Supplementary Data 4). **b** UMAP visualization of expression of LGE-specific marker genes, GSX2, SALL3, RP11-849l19.1, MEIS2, PBX3, and FOXP2 in 1-month-old organoids. **c** Organoid sections immunostained for PAX6 and GSX2. **d** Quantification of percentages of PAX6-, GSX2- and PAX6/GSX2-expressing cells in the lumen-surrounding VZ (*n* = 7 organoids/1–2 sections per organoid in two batches from 2242-5 and EP2-15 lines). **e** Angular distributions of PAX6-, GSX2- and PAX6/GSX2-expressing cells in the VZ (*n* = 3 organoids/1 sections per organoid from 2242-5 line). **f**, **g** Images of fetal human brain tissue sections at 9 (**f**) and 11 (**g**) PCW stained using FISH probes against *PAX6* and *GSX2* (**f**) or immunostained with antibodies against PAX6 and GSX2 (**g**). **h** Organoid sections immunostained for CTIP2 and GABA. **i** Percentage of inhibitory neurons (GABA- or GAD67 expressing cells) that co-expressed CTIP2 (*n* = 11 organoids/1–5 sections per organoid in 3 batches from H9 and 2242–5 lines). Data are means ± s.e.m. Scale bars = 100 and 50 (zoom-in) (**c**, **h**), 500 (**f**), and 200 (**f**, **g**) μm. Source data are provided as a Source data file.

To characterize the reproducibility and cellular composition in 5-month-old organoids, we performed scRNA-seq and unsupervised clustering on 22,486 single cells collected from six organoids that were produced in separate batches from four different PSC lines (Fig. 5g). All cells were clustered in 14 molecularly distinct cell clusters (Fig. 5i, Supplementary Fig. 9, and Supplementary Data 5), and each cluster was composed of cells collected from multiple organoids (Fig. 5h), indicating the reproducibility of cellular composition. To compare the reproducibility of cellular composition in SNR-derived organoids and organoids produced using other protocols, as well as primary human and mouse brain tissue from the previous studies[35,36], we used mutual information (MI) and z-scores that were calculated on scRNA-seq

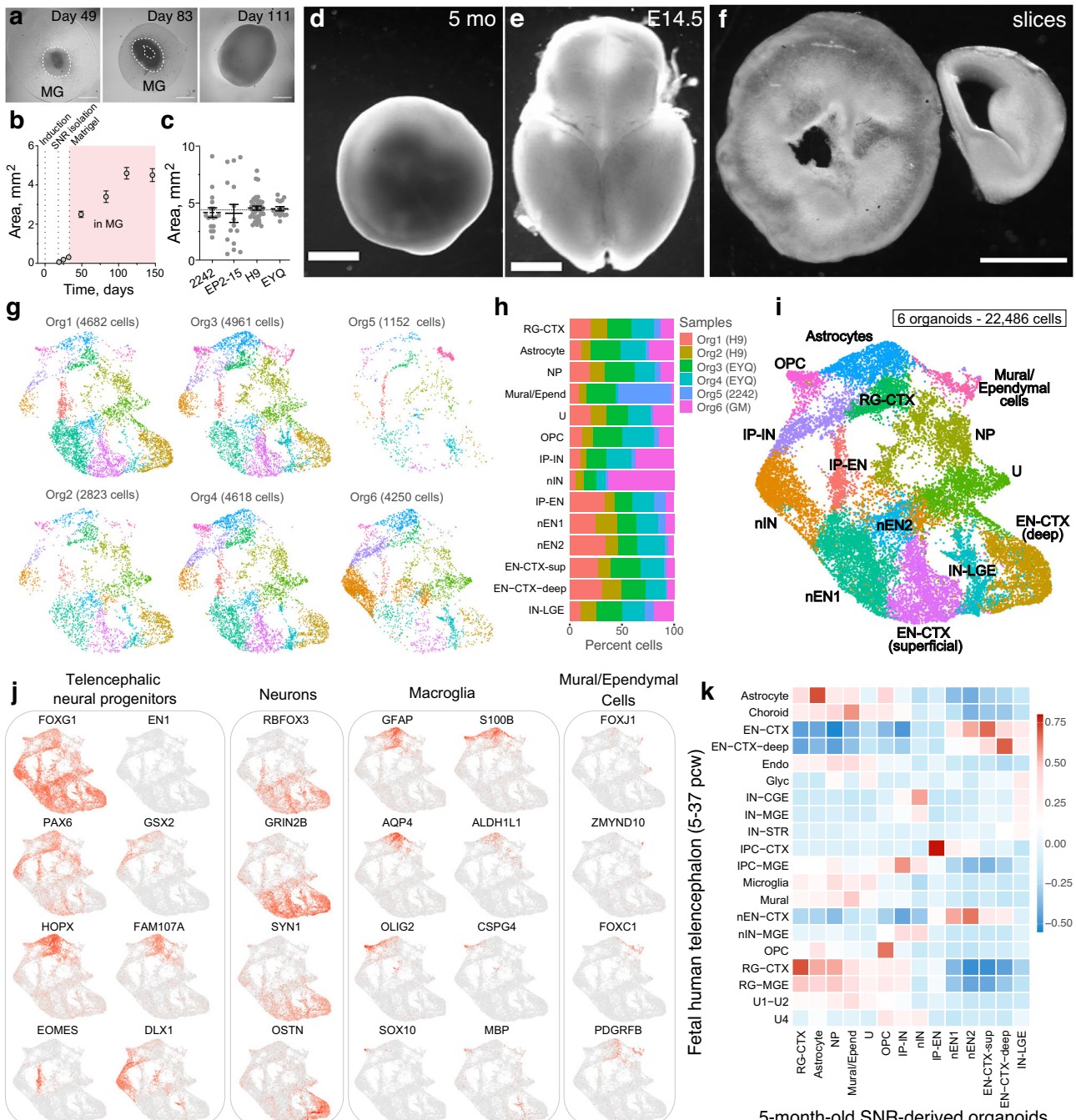

**Fig. 5 | Reproducibility and diversity of telencephalic cells in 5-month-old SNR-derived organoids. a** Images of SNR-derived organoids at day 49, 83, and 111 post SNR isolation. **b** Quantification of organoid sizes over time (*n* = 47 [day19], 46 [day25], 22 [day32], 25 [day49], 26 [day83], 26 [day111] and 25 [day146] organoids produced in two differentiation batches from H9 line). **c** Quantification of sizes of 5-month-old SNR-derived organoids produced from different stem cell lines (*n* = 17, 15, 38, 15 organoids produced in 2–4 differentiation batches from 2242-5, EP2-15, H9, and EYQ2-20 lines, respectively). **d–f** Image of a 5-month-old SNR-derived organoid (**d**) and 350-µm-thick organoid slice (**f**) in comparison with E14.5 mouse brain (**e**) and coronal mouse brain hemisphere slice (**f**). **g** UMAP visualizations of

unsupervised clustering of cells obtained from 5-month-old SNR-derived organoids generated from different stem cell lines. The clustering was performed on the combined dataset. **h** Percentages of cells from different organoids in different clusters. **i** UMAP visualization of combined 22,486 cells from six organoids. **j** Heatmap visualizations of expression of selected cell type-specific genes. **k** Heatmap of Pearson correlation between cell clusters identified in 5-month-old organoids and those in primary human fetal tissue from Nowakowski et al.[7]. Data are mean ± s.e.m. Scale bars = 1 mm. RG radial glia, IP intermediate neural progenitor, EN excitatory neuron, IN inhibitory neuron, CTX cortex, OPC oligodendrocyte progenitor cell, U unknown. Source data are provided as a Source Data file.

datasets processed using the same clustering parameters (Supplementary Fig. 12). This analysis demonstrated that the reproducibility in 1- and 5-month-old SNR-derived organoids is comparable to that in the dorsal forebrain organoids generated using a directed differentiation protocol with a Wnt inhibitor (IWR1) and to that in primary human and

mouse cortical tissue samples[36]. We further confirmed this result using bulk RNA sequencing and correlation analysis (Supplementary Fig. 13), which is less sensitive for assessing the reproducibility of cellular composition but have frequently been used in the previous studies on organoids[17,18,22,54] and human fetal brain tissue[55]. Together, these results

indicate that the composition of cells in SNR-derived organoids is relatively consistent and reproducible.

As in 1-month-old organoids, most cells in 5-month-old organoids showed ubiquitous expression of the telencephalic marker *FOXG1* (Fig. 5j) and no detectable expression of the endodermal, mesodermal, or posterior ectodermal markers (http://organoid.chpc.utah.edu), suggesting the telencephalic identity. In contrast to 1-month-old organoids, 5-month-old organoids contained cell clusters with increased expression of the oRG markers *HOPX, FAM107A, TNC, PTPRZ1, LIFR,* and *MOXD1*; mature neuronal markers *RBFOX3* (NeuN), *GRIN2B, SYN1,* and *OSTN*[65]; macroglial markers *GFAP, S100B, AQP4, ALDH1L1, OLIG2, CSPG4* (NG2), *SOX10,* and *MBP*; ependymal cell markers *FOXJ1, ZMYND10,* and *S100B*; and periendothelial cells markers *FOXC1, PDGFRB,* and *ACTA2* (Fig. 5j and http://organoid.chpc.utah.edu). Importantly, the gene expression profiles detected in cells from 5-month-old SNR-derived organoids were similar to those detected in cells from the fetal human telencephalon[7] (Fig. 5k) at mid-fetal developmental stages (12–24 PCW) (Supplementary Fig. 14).

To characterize the organization of 5-month-old organoids and distributions of detected cells, we performed immunostaining with cell-type specific markers (Fig. 6). We first confirmed that there was no enrichment of apoptotic, cleaved Cas3-expressing, cells near the lumen area in organoid sections (Fig. 6a, b) and that most cells from dissociated organoids were viable (Trypan blue negative: 90 ± 1.7%, *n* = 7 organoids). Instead, we found that the lumen was surrounded by a layer of PAX6-expressing NPs (-10% of all cells) (Fig. 6c, e) or lined by S100β- and FOXJ1-expressing ependymal-like cells (Fig. 6f1). These results are consistent with the idea that the lumen in 5-month-old SNR-derived organoids is likely a ventricle-like structure derived from SNR and not the area of excessive apoptosis and cell death. We also observed that neurons, characterized by the expression of TUJ1 or MAP2 (-35% of all cells, Fig. 6e), were radially oriented and positioned more basally relative to the lumen (Figs. 6d). Interestingly, GFAP-expressing cells (-40% of all cells, Fig. 6e) were also positioned basally to the lumen and exhibited oRG- (Fig. 6d1) or astrocyte-like morphologies (Fig. 6d2). Other cells detected in 5-month-old SNR-derived organoids, including S100β-expressing astrocytes (Fig. 6f2), O4- and MBP-expressing oligodendrocytes (<1% of all cells, Fig. 6e) (Fig. 6g), and PDGFRβ- and α-SMA-expressing periendothelial cells (Fig. 6h) were distributed throughout the organoid sections. We found no evidence of myelination in 5-month-old SNR-derived organoids. These results suggest that multiple neural cell types, including NPs, oRGs, neurons, astrocytes, oligodendrocytes, and ependymal and periendothelial cells, are generated in 5-month-old SNR-derived organoids and that these cells have predictable organization relative to the lumen.

## Cortical and striatal neurons in SNR-derived organoids

We next investigated the diversity and distribution of neurons in 5-month-old SNR-derived organoids using scRNA-seq and immunostainings (Fig. 7). Cell in the neuronal clusters (nEN1, nEN2, EN-CTX [deep], EN-CTX [superficial], and IN-LGE) (Fig. 5i, j) were characterized by the expression of either EN marker *SLC17A7* (VGLUT1) or IN markers *GAD1* and *GAD2* (Fig. 7a). ENs demonstrated increased expression of the typical deep layer cortical markers *FEZF2* and *LMO3* or superficial layer cortical markers *SATB2, CUX2,* and *MDGA1*; while INs were characterized by the expression of ISPN markers *PBX3, FOXP2, EBF1,* and *SIX3* or/and interneuron markers *SST, NPY, VIP CALB1, CALB2,* and *NOS1* (Fig. 7a). Both ISPNs and interneurons were predominantly distributed to the IN-LGE cluster, suggesting the similar origin of these cells. Interestingly, we also found a small proportion of *NKX2-1-* and *LHX6-*expressing cells in the IP-IN and nIN clusters, as well as Sonic Hedgehog (SHH) expressing neurons in the IN-LGE and EN-CTX-Deep clusters (http://organoid.chpc.utah.edu), suggesting that a small proportion of MGE-derived NPs is specified in 5-month-old SNR-derived organoids.

In the sections, cortical ENs showed a laminar organization (Fig. 7b, c and Supplementary Fig. 15), with neurons expressing deep-layer cortical markers TBR1 or CTIP2 (20–40% of neurons [Fig. 7d]) laying beneath those expressing superficial-layer cortical marker SATB2 (-10% of neurons [Fig. 7d]). No layer-specific expression was observed for CUX1 (Fig. 7c and Supplementary Fig. 15), which is consistent with its non-specific expression pattern in organoid clusters (http://organoid.chpc.utah.edu) and fetal human cortical tissue[7]. INs expressing inhibitory interneuron markers GABA or GAD67 and calretinin (-4% of GABA-expressing cells), calbindin (-5%), somatostatin (-7%), parvalbumin (-4%), or vasoactive intestinal polypeptide (-4%), were distributed throughout the tissue (Fig. 7e, f). Interestingly, INs, expressing ISPN markers GABA, CTIP2, and DARPP32 (-3% of GABA-expressing cells), were predominantly found beneath the cortical layers characterized by DARPP32 or/and CTIP2 (Fig. 8a–d, and Supplementary Fig. 16). To validate the presence of ISPNs in 5-month-old organoids, we performed patch-clamp electrophysiology, biocytin filling, and post-hoc immunostaining for ISPNs markers GAD67 and FOXP2[8,56] for functional-, morphological-, and immunostaining-based identification of these cells (Fig. 8e–i and Supplementary Fig. 17). Although -25% of GAD67-expressing cells demonstrated nuclear FOXP2 expression in organoid slices (Supplementary Fig. 17), only one such cell was found among 34 recorded and biocytin-filled cells (Fig. 8e–i). This GAD67- and FOXP2-expressing neuron was characterized by an increased membrane capacitance (100 pF), as compared to all recorded cells ($C_{median}$ = 56 pF, CI95% = 40 and 66 pF), more negative resting membrane potential (−71.4 mV vs −50.9 mV, CI95% = −57.4 and −43.4 mV), increased latency of the initial action potential (AP) (166 ms vs 75 ms, CI95% = 62 and 112 ms) and lack of sag and spike-frequency adaptation (Fig. 8h, i), which are the distinguishing functional characteristics of ISPNs[57–59]. In addition, the distal dendrites of this neuron were covered with dendritic spines (Fig. 8f), which is a distinguishing morphological feature of ISPNs in the brain. To explore why ISPNs have not been observed in some of the previous studies on telencephalic organoids[25,29,35,36,60], we investigated the expression profiles of key cell identity markers, including FOXG1, PAX6, and GSX2, in those organoids (Supplementary Fig. 18). Interestingly, we found no GSX2-expressing NPs, suggesting that GSX2 expression is likely important for the speciation of ISPNs in SNR-derived organoids observed in this study. Together, these results support the idea that cortical ENs and inhibitory ISPNs and interneurons are generated in 5-month-old SNR-derived organoids.

## Functional neurons and neural networks in SNR organoids

To further investigate the functional properties of neurons in 5-month-old SNR-derived organoids, we performed patch-clamp electrophysiology on the cells located in the layered slice regions (Fig. 9a) that demonstrated pyramidal neuron-like morphologies (Fig. 9b). Overall, we found that -86% (103/120) of recorded cells fired action potentials (APs) and generated diverse patterns of electrical activity (Fig. 9c, Supplementary Data 6, and http://organoid.chpc.utah.edu). To characterize the diversity of functional neurons in 5-month-old SNR-derived organoids, we extracted multiple non-overlapping electrophysiological characteristics from the recordings of voltage deflections induced in response to different somatic current injections and performed unsupervised clustering (Fig. 9d). The recorded cells could be divided into five groups largely based on their differences in passive membrane properties (input resistance, resting membrane potential, and capacitance), single action potential (AP) profile (AP amplitude, latency, threshold, width, after-hyperpolarization potential amplitude [AHP], and sag) and AP firing pattern (frequency and adaptation) (Fig. 9c, d). This is consistent with the idea that neurons with diverse functional responses are generated in 5-month-old SNR-derived organoids.

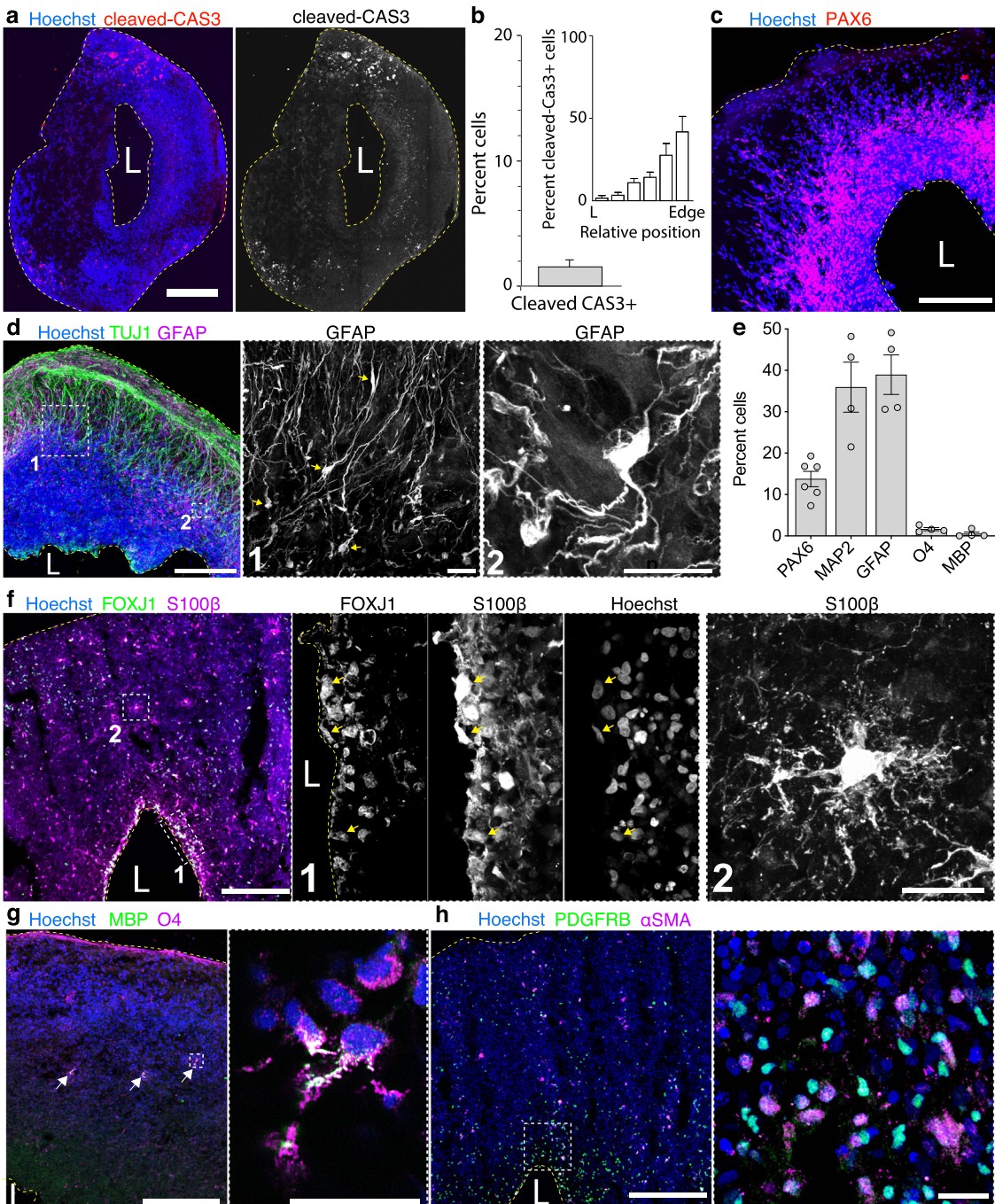

**Fig. 6 | Diversity of neural cells, including outer radial glia, astrocytes, and ependymal- and periendothelial-like cells are generated in 5-month-old SNR-derived organoids. a** Organoid section immunostained for cleaved caspase-3. **b** Percentage and distribution of cleaved caspase 3-expressing cells in organoid sections (*n* = 4 sections from two organoids produced in two batches from 2242-5 and H9 lines). **c**, **d**, **f**–**h** Organoid sections immunostained for PAX6 (**c**), TUJ1 and GFAP (**d**), FOXJ1 and S100β (**f**), O4 and MBP (**g**), and PDGFR-β and α-SMA (**h**). **e** Percentages of cells expressing cell type-specific markers (*n* = 6 [PAX6], 4 [MAP2], 4 [GFAP], 4 [O4] and 4 [MBP] organoids/1–3 sections per organoid produced in 1–3 batches from 2242-5, EP2-15, and H9 lines). Data are mean ± s.e.m. Scale bars = 500 (**a**), 200 and 20 (zoom-in) (**c**, **d**, **f**–**h**) μm. L lumen. Source data are provided as a Source Data file.

We next investigate the synaptic properties of neurons in 5-month-old SNR-derived organoids via synaptic immunostaining and electrophysiology (Fig. 9e–i). Interestingly, the density of synaptic puncta in organoid sections, visualized by immunostaining for a pre-synaptic marker BASSOON, increased along the lumen to the periphery axis (Fig. 9e). This change in synaptic density is consistent with an inside-out organization of NPs and neurons in organoids. Co-immunostaining with the pre- and postsynaptic markers, including BASSOON/VGLUT1, BASSOON/SHANK3, VGLUT1/PSD95, BASSOON/ HOMER1, BASSOON/SHANK2, BASSOON/SHANK1 and BASSOON/ Gephyrin, revealed the presence of different subtypes of putative excitatory and inhibitory synapses (Fig. 9f, g). Consistent with this observation, both spontaneous excitatory and inhibitory postsynaptic currents (EPSCs and IPSCs, respectively) were detected in ~56% of recorded cells (9 out of 16) (Fig. 9h, i and Supplementary Data 6). EPSCs were characterized by fast rise and decay kinetics and could be completely blocked by application of the AMPA and NMDA receptor antagonist, kynurenic acid (Fig. 9h). IPSCs showed significantly slower

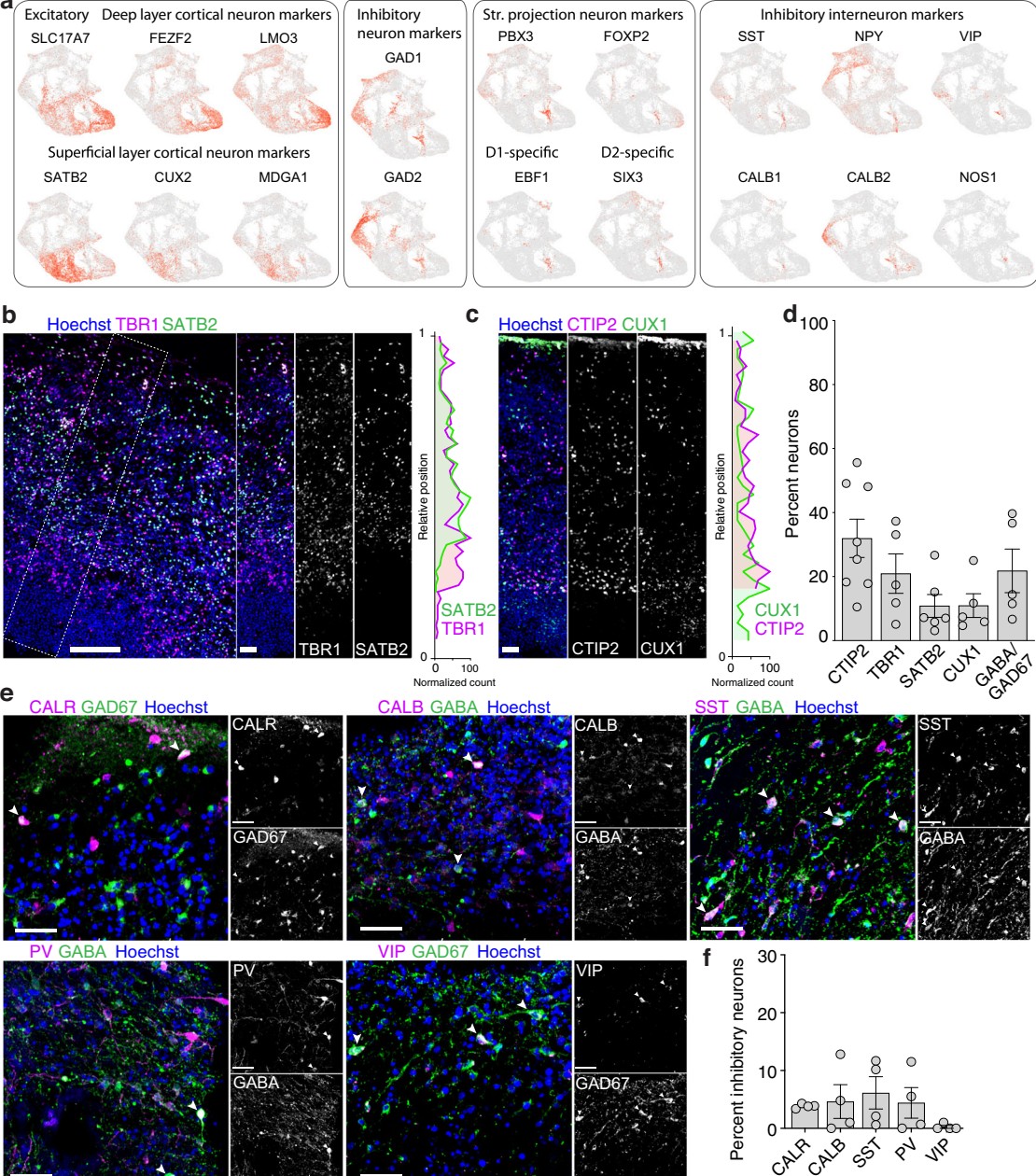

**Fig. 7 | Characterization of cortical excitatory neurons and inhibitory interneurons in 5-month-old SNR-derived organoids. a** Heatmap visualizations of selected neuronal markers. **b, c** Organoid sections immunostained for TBR1, and SATB2 (**b**) and CTIP2, and CUX1 (**c**) and relative distributions of cells expressing these markers the marked ROIs. **d** Percentages of cells expressing cell different neuron type-specific markers (*n* = 8 [CTIP2], 5 [TBR1], 6 [SATB2], 5 [CUX1] and 5 [GABA/GAD67] organoids/1–2 sections per organoid produced in 2–3 batches from 2242-5 and EP2-15 lines). **e** Organoid sections immunostained for Calretinin (CALR) and GAD67; Calbindin (CALB) and GABA; Somatostatin (SST) and GABA; Parvalbumin (PV) and GABA; and Vasoactive intestinal peptide (VIP) and GAD67. **f** Percentages of inhibitory neurons expressing different interneuron markers (*n* = 4 organoids/1–3 sections per organoid, produced in four batches from 2242-5, EP2-15, H9, and EYQ2-20 lines). Data are presented as mean ± s.e.m. Scale bars = 200 and 20 (zoom-in) (**b, c**) and 50 (**e**) μm. Source data are provided as a Source Data file.

kinetics and could be completely abolished by application of a GABA-A receptor antagonist, picrotoxin (Fig. 9i). Together, these results demonstrated that a proportion of neurons in 5-month-old SNR-derived organoids are functionally mature and interconnected through both excitatory and inhibitory synapses.

The detection of organized and functionally mature neurons in 5-month-old SNR-derived organoids encouraged us to investigate functional neural networks that has been observed in some of the previous studies on organoids[60–62]. For this, 5-month-old SNR-derived organoids were implanted with flexible miniature 16-channel multi-

electrode arrays (Fig. 9j). We observed clear patterns of spontaneous electrophysiological activity around 1-month post-implantation (Fig. 9k) with some waveforms resembling typical extracellularly recorded APs (Fig. 9l). To investigate functional connections, we electrically stimulated one of the active electrodes and recorded the responses from all electrodes (Fig. 9m). We found that relatively strong stimuli (100-mA bursts) elicited responses in multiple channels adjacent to the stimulation electrode; while weaker stimuli (1-mA single pulse) did not reach the threshold for inducing electrical responses in the adjacent channels. These results indicate the presence of maturing

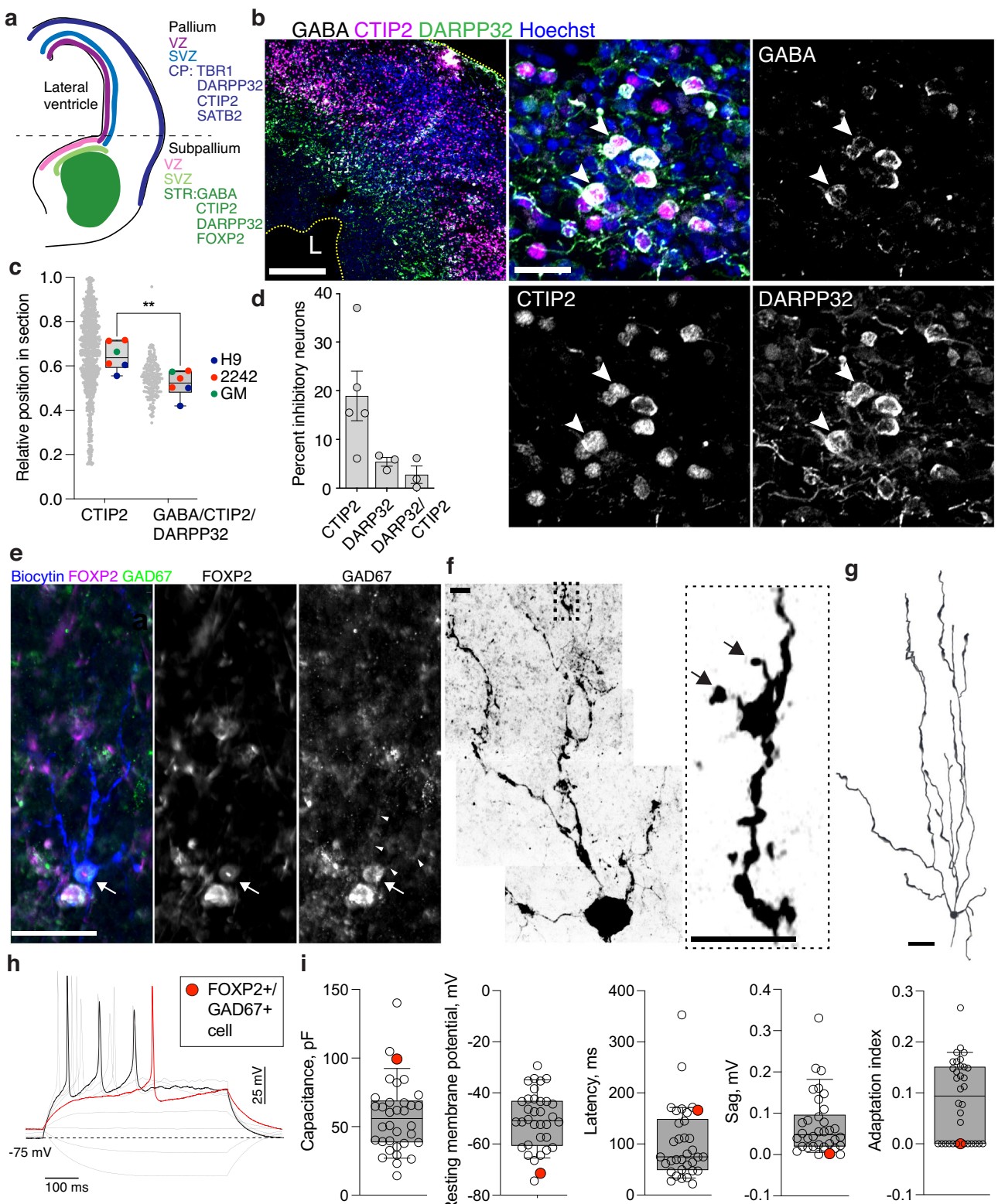

functional neural connections in 6-month-old SNR-derived organoids. The activity was abolished by bath application of a sodium channel antagonist, tetrodotoxin (TTX) (Fig. 9n), supporting the AP-dependent nature of detected electrical activity in organoids. To investigate the presence of oscillatory rhythms, we performed a spectral analysis of the recorded signals, before and after TTX administration (Fig. 9o, p). Oscillations of different frequencies were detected in SNR-derived organoids with an average rate of 2.5 Hz, which substantially exceeded the noise level, and were abolished upon TTX application. No peaks were detected in the alpha (8–12 Hz) or beta (15–25 Hz) frequency bands, which are associated with the activity of thalamic neurons and thalamo-cortical neural networks. Together, these results indicate the presence of functional neural networks in 6-month-old SNR-derived organoids.

**Fig. 8 | Characterization of inhibitory striatal projection neurons in 5-month-old SNR-derived organoids. a**, Cartoon depicting expression patterns of neuronal marker genes in fetal human telencephalon. **b** Organoid sections immunostained for GABA, CTIP2, and DARPP32. **c** Relative distributions of CTIP2- and CTIP2/DARPP32/GABA -expressing cells in organoid section (*n* = 6 organoids/1–3 sections per organoid produced in three differentiation batches from 2242-5, H9, and GM07492 lines). **d** Percentages of inhibitory neurons expressing striatal neuron markers (*n* = 5 [CTIP2] and 3 [DARPP32 and CTIP2] organoids/1–3 sections per organoid, produced in four batches from 2242-5, EP2-15, and H9 lines). **e–g** Image of GAD67- and FOXP2-expressing neuron (**e**) that was recorded and filled with biocytin to visualize dendritic spines (**f**) and morphology (**g**). **h** Recordings of voltage deflections obtained in response to Δ10pA somatic current injections from neuron depicted in (**e**). **i** Measurements of membrane capacitance, resting membrane potential, latency of 1st action potential, sag amplitude, and action potential adaptation index obtained from recorded cells (*n* = 34 cells recorded from four organoids obtained in two differentiation batches from 2242-5 and EP2-15 lines). The measurements obtained from neuron depicted in (**e**) are in red. Data are presented as means ± s.e.m. (**d**), individual data point and box plots (median ± 25th/75th [box] and min and max values [whiskers]) (**c**), and individual data points and box plots (median ± 25th/75th [box] and 10th/90th percentiles [whiskers]) (**i**). Scale bars = 200 and 20 (zoom-in) (**b**), 50 (**e**, **g**), and 5 (**f**). Source data are provided as a Source Data file.

## Cellular and molecular deficits in SHANK3-deficient organoids

To test whether SNR-derived organoids can be used to model genetic neurodevelopmental disorders, we generated CRISPR/Cas9-engineered PSC lines with partial homozygous or complete hemizygous *SHANK3* deletions (Fig. 10a, b). The incomplete homozygous *SHANK3* deletion (*SHANK3*−/−) resulted in complete loss of expression of all SHANK3 isoforms in PSC-derived neurons (Supplementary Fig. 19), whereas complete hemizygous *SHANK3* deletion (*SHANK3*+/−), which is the most common genetic abnormality detected in Phelan-McDermid syndrome patients[63], resulted in ~50% loss of expression of the longest SHANK3 isoform[64].

We first examined the size, overall self-organization, and neuronal/NP composition in isogenic control (iCtrl) and SHANK3-deficient organoids. While all organoids exhibited similar lumen-based self-organization and increased in size over time (Supplementary Fig. 5 and Fig. 5c), *SHANK3*-/- (EP2-15), but not *SHANK3*+/− (EYQ2-20), organoids, were significantly smaller than respective iCtrl organoids (2242-5) at 1-month post-induction (Supplementary Fig. 5f). In addition, we found that *SHANK3*-/- organoids contained reduced proportions of neurons with smaller nuclei sizes than iCtrl organoids (Supplementary Fig. 20a–d), suggesting the size phenotype detected in *SHANK3*-/- organoids could be attributed to reduced proportions and sizes of neurons. However, the size phenotype is unlikely to be relevant to PMDS patients as it was not observed in *SHANK3*+/− organoids at 1- (Supplementary Fig. 5f) or 5-months (Fig. 5c), and no differences were detected in the proportion or sizes of neurons in iCtrl and *SHANK3*+/− organoids (Supplementary Fig. 20e–g).

We next investigated the proportions of SHANK3-containing synaptic puncta in 5-month-old control and SHANK3-deficient organoids (Fig. 10d, e). Both *SHANK3*-/- and *SHANK3*+/− organoids demonstrated significantly reduced proportions of SYNAPSIN1/SHANK3-containig excitatory synaptic puncta as compared to the respective iCtrl organoids (Fig. 10e). This result is consistent with the results of the previous studies on SHANK3-deficient human neurons with different deletions and mutations[65–68]. It also supports the idea that the longest *SHANK3* isoform is an essential regulator of excitatory synapse development in human neurons so that even a partial reduction in the expression level of this isoform causes excitatory synaptic deficits in human telencephalic tissue.

We further examined the intrinsic excitability of neurons in iCtrl and SHANK3-deficient organoids by measuring voltage deflections in response to different somatic current injections using whole-cell patch-clamp electrophysiology (Fig. 10f, g and Supplementary Fig. 21). SHANK3-deficient neurons were more excitable than iCtrl neurons (Fig. 10g). Specifically, both *SHANK3*-/- and *SHANK3*+/− neurons demonstrated a shorter AP latency as compared to respective iCtrl neurons (Fig. 10g). In addition, *SHANK3*-/- neurons, but not *SHANK3*+/−, were characterized by a smaller membrane capacitance, more hyperpolarized threshold for AP generation, reduced amplitude of after-hyperpolarization potential (AHP), and increased firing rate as compared to iCtrl neurons (Fig. 10g). Collectively, these results are consistent with the idea that SHANK3 is an important regulator of intrinsic excitability in human neurons and its loss leads to intrinsic neuronal hyperexcitability[65,66].

Although SHANK3 is the main candidate gene for the developmental deficits in PMDS patients, the role of SHANK3 in regulating the early aspects of human telencephalic development, including proliferation, neurogenesis, and neural cell type specification, which may be disrupted in ASD-associated developmental disorders[34], remain unclear. We compared the proportions of different telencephalic cell types in iCtrl and *SHANK3*+/− organoids (Fig. 10h, i). Interestingly, we found no major differences in the cellular composition of different cell clusters (Fig. 10h) and expression levels of cell type specific markers between iCtrl and *SHANK3*+/− organoids (Fig. 10i). These results suggest that *SHANK3* hemizygosity is unlikely to disrupt early neurodevelopmental processes in the human telencephalon.

To gain insights into the molecular pathways that are disrupted by *SHANK3* hemizygosity in human telencephalic development, we performed bulk RNA-seq on another three pairs of 5-month-old iCtrl and *SHANK3*+/− organoids (Fig. 10j and Supplementary Data 7). Interestingly, we found several clustered protocadherins and the cadherin signaling pathways (P00012) downregulated in *SHANK3*+/− organoids as compared to iCtrl organoids. To validate this observation, we performed bulk RNA-seq on another three pairs of 5-month-old organoids generated from an independent control iPSC line (GM07492) and SHANK3-defient iPSC line from a patient with a partial *SHANK3* deletion (Supplementary Fig. 22). Analysis of DEGs in 6 control and 6 *SHANK3*+/− organoids identified significantly reduced expression of several clustered protocadherins, including *PCDHA6*, *PCDHA7*, and *PCDHA9*, and enrichment in the cadherin signaling pathway (P00012) (Fig. 10k, l). Collectively, these results indicate that *SHANK3* hemizygosity in human telencephalic tissue is associated with impaired expression of clustered protocadherins, which has previously been implicated in regulating neural circuit formation[69–71] and identified mutated in individuals with autism[72].

## Discussion

The present study describes a method for generating human telencephalic organoids from stem cell-derived SNRs. The use of SNRs for generating organoids offers several potential advantages. First, it mimics the development of neural tissue from a singular neural tube in vivo. Previous studies reported the generation of organoids or spheroids with multiple neural rosettes from pluripotent or multipotent neural stem cells[13–17,20,26,32,60]. However, the presence of multiple rosettes per organoid results in an unpredictable number of germinal zones and tissue organization within an organoid. Moreover, neural tissue with multiple rosettes-like structures has been observed in brain tumors[73–75], which may confound the use of organoids for modeling normal development and disease. Second, SNR-derived organoids showed a relatively consistent and reproducible composition of different telencephalic cell types, which is similar to the results of other recent studies on organoids produced from single neural rosettes[76–78]. This allowed us to compare and contrast the cellular and molecular properties of control and isogenic SHANK3-deficient organoids to

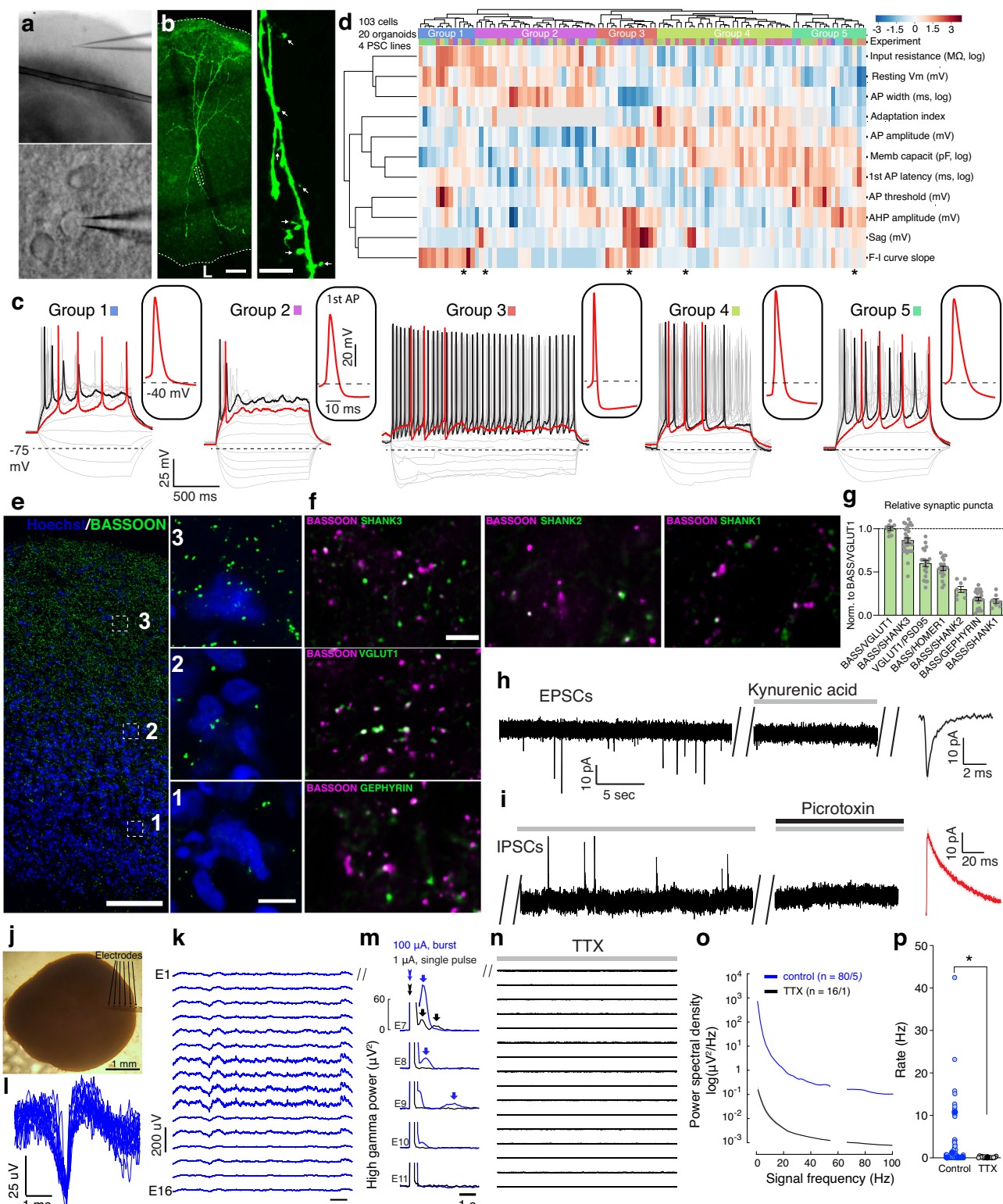

identify differentially expressed genes with no major differences in cellular composition. Finally, SNR-derived organoids contain functionally mature neurons and synapses. This allowed us to use these organoids to identify intrinsic and synaptic deficits caused by SHANK3 deficiency in human telencephalic tissue. Collectively, our results demonstrate that SNR-derived organoids recapitulate early aspects of human telencephalic development and could be used to model it under normal and pathological conditions.

We showed that different types of telencephalic NPs, cortical and striatal neurons, as well as astrocytes, oligodendrocytes, ependymal cells, and periendothelial cells are generated in SNR-derived organoids in a timely manner. Interestingly, ependymal cells and periendothelial cells, such as pericytes and vascular smooth muscle cells, have not been reported in previous studies of organoids. These cells are the essential components of the brain barriers[79]; however, their developmental origins and molecular properties in the developing human

**Fig. 9 | Functional neurons and neural networks in 5-month-old SNR-derived organoids. a** Low- (top) and high- (bottom) resolution images of a 5-month-old organoid slice used for electrophysiology. **b** Image of a biocytin-filled control neuron with dendritic spines in organoid slice after electrophysiology. **c** Traces of membrane potentials recorded from cells (marked with asterisk in **d**) in different clusters in response to Δ5 pA somatic current injections. **d** Heatmap of unsupervised hierarchical clustering of recorded neurons based on non-redundant electrophysiological characteristics ($n = 103$ cells from 4 2242-5, 3 EP2-15, 11 H9, and 2 EYQ organoids). **e, f** Organoid section immunostained for presynaptic marker BASSOON. **e** BASSOON and SHANK3, SHANK2, SHANK1, VGLUT1, or GEPHYRIN (**f**). **g** Quantification of excitatory and inhibitory synaptic puncta ($n = 8$ [Bassoon:V-GluT1], 28 [Bassoon:SHANK3], 20 [VGluT1:PSD-95], 20 [Bassoon:Homer], 8 [Bassoon:SHANK2], 24 [Bassoon:Gephyrin], and 8 [Bassoon:SHANK1] sections from 1–3 organoids produced in one batch from 2242-5 line). **h, i** Traces of spontaneous excitatory postsynaptic currents (EPSCs) (**h**) and inhibitory postsynaptic currents

(IPSCs) (**i**) recorded at different holding membrane potentials without inhibitors and with 3 mM kynurenic acid (glutamate receptor antagonist) or 100 μM picrotoxin (GABA receptor antagonist). **j** Image of a 5-month-old SNR-derived organoid implanted with a 16-channel multi-electrode array. **k–n** Spontaneous electrical activity recorded from different electrodes (E1–16) before (**k**) and after tetrodotoxin (TTX, 1 μM) application (**n**). Waveforms extracted after filtering of low-frequency components with a 300-Hz high-pass filter (**l**). Extracellular potentials recorded from different electrodes after delivery of stimuli of varying intensity at E7 (**m**). **o, p** Spectral analysis of recorded signals (**o**) and average rate of high-frequency events (>300 Hz) before ($n = 80$ fields obtained from five organoids [16 fields per organoid] produced in one differentiation batch from H9 line) and after TTX application ($n = 16$ fields obtained from 1 H9 organoid) (**p**). Data are mean ± s.e.m.; $*P = 0.047$, unpaired two-sided Kolmogorov-Smirnov test. Scale bars = 50 and 10 (zoom in) (**b**), 200 and 20 (zoom in) (**e**), and 20 and 10 (zoom in) (**f**) μm. Source data are provided as a Source Data file.

brain are not well-understood[80,81]. In SNR-derived organoids, both cell types likely originated from NPs, as no mesodermal, endodermal, or neural crest cells were detected, and both showed increased expression of genes encoding for components of the extracellular matrix, such collagen and fibronectin, and secreted peptides, such as IGF2 and RSPO2. The specific types of progenitors and molecular programs that are responsible for the production of these cells in SNR-derived organoids remain to be determined.

We also demonstrated that different types of INs are generated in SNR-derived organoids, including interneurons, characterized by the expression of the typical interneuron markers SST, PV, VIP, CALR, and CALB, and ISPNs, characterized by the co-expression of GABA and CTIP2, GABA, CTIP2, and DARRP32, or GAD67 and FOXP2. Although INs have been reported in several previous studies on telencephalic and cortical organoids[17,35,36], the origin of these cells and transcriptional programs responsible for their specification have not been investigated. Our results indicate that both IN and EN lineages in SNR-derived organoids are likely generated from a common pool of NPs that was characterized by the expression of early NP markers PROM1 and NES and the Wnt pathway associated genes *RSPO1, GPC3*, and *FZD5*. The specification of IN lineage was associated with a decreased expression of the Wnt pathway associated genes and increased expression of multiple genes associated with IN development, including *GSX2, SALL3, ASCL1*, and *DLX1/2*. GSX2 is an important homeobox transcription factor required for the specification of LGE NPs in the mouse brain[82]. We found GSX2 expression restricted to the LGE and pallial-subpallial boundary (PSB) in the developing human telencephalon (Fig. 4f, g). This suggests that the majority of INs in SNR-derived organoids are likely of the LGE/PSB origin. Consistent with this idea, an increased expression of the LGE-specific marker genes was detected in the NP4 and nIN3-5 cell clusters and regionalized LGE-like distribution of GSX2-expressing cells was observed in 1-month-old organoids. In the rodent brain, LGE/PSB NPs contribute to the production of a diverse population of INs, including IPSNs[83–85] and olfactory bulb inhibitory interneurons[86]. In the primate brains, LGE NPs also contribute to the production of cortical inhibitory interneurons[87,88]. This is consistent with the idea that a mixed population of telencephalic LGE-derived INs, including cortical and olfactory bulb interneurons and ISPNs, is likely generated in SNR-derived organoids. The specification of both pallial and subpallial LGE-derived cells in SNR-derived organoids was potentially achieved as a result of using a vitamin A-containing differentiation media. Indeed, it has been demonstrated that retinoic acid, an immediate derivative of vitamin A, is an important regulator of both cortical and striatal development[89–92] and that striatal organoids can be generated by using an agonist of retinoid X receptors, SR11237[30]. As both cortical ENs and striatal INs are produced in SNR-derived organoids, in the future, it will be important to investigate how these neurons

communicate with each other and whether they form predominantly unidirectional cortico-striatal connections, as it was observed using cortico-striatal assembloids[30]. As a first step in this direction, we characterized the intrinsic properties of neurons in SNR-derived organoids and confirmed the presence of both excitatory and inhibitory synapses. In addition, we recorded both low- (4–30 Hz) and mid- to high-frequency (30–100 Hz) oscillations. However, it remains to be determined how different ENs and INs contribute to activity in SNR-derived organoids and whether and how this activity evolves over time.

Finally, in this study, we identified the cellular, molecular, and functional deficits in human telencephalic organoids with a complete hemizygous *SHANK3*+/− deletion, which has been detected in most patients with Phelan-McDermid Syndrome[93,94]. Interestingly, we observed no obvious cell identity deficits in *SHANK3*+/− organoids, suggesting that the molecular mechanisms regulating the specification and identity of neural cells in telencephalic tissue are unlikely to be affected by *SHANK3* deletion in patients. However, we found that *SHANK3*+/− organoids have reduced numbers of excitatory synapses and contain neurons with elevated intrinsic excitability. These results are consistent with the results of the previous studies on patient iPSC-derived neurons with large 22q13 deletions encompassing multiple genes[65] and engineered induced human neurons with a partial intragenic *SHANK3* deletion[66], suggesting that the expression level of the longest *SHANK3* isoform is an essential regulator of neuronal excitability and excitatory synapses in human tissue. To gain insights into the molecular pathways that may contribute to the development of these deficits, we investigated the differentially expressed genes in control and SHANK3-deficient organoids generated from patient and engineered stem cells lines with hemizygous *SHANK3*+/− deletion. We discovered that SHANK3-deficient organoids exhibited reduced expression of multiple clustered protocadherins as compared to control organoids. The clustered protocadherins are stochastically expressed in individual neural cells[95] and important for proper brain wiring[71]. Similarly to *SHANK3*, clustered protocadherins have been implicated in controlling neural morphogenesis and synaptogenesis under normal and pathological conditions[70,96,97]. In the future, it will be important to elucidate the mechanistic link between hemizygous *SHANK3* deletion, reduced expression of clustered protocadherins, and functional deficits detected in SHANK3+/− neurons and organoids. In addition, the detected deficits will need to be confirmed in organoids from more patient and engineered iPSC lines as it was technically challenging to correct the large chromosomal abnormality in patient iPSC line and to generate additional isogenic SHANK3-deficient lines with a complete hemizygous *SHANK3* deletion.

In summary, this study validates SNR-derived organoids as a reliable model for studying human telencephalic development under normal and pathological conditions, provides insights into the

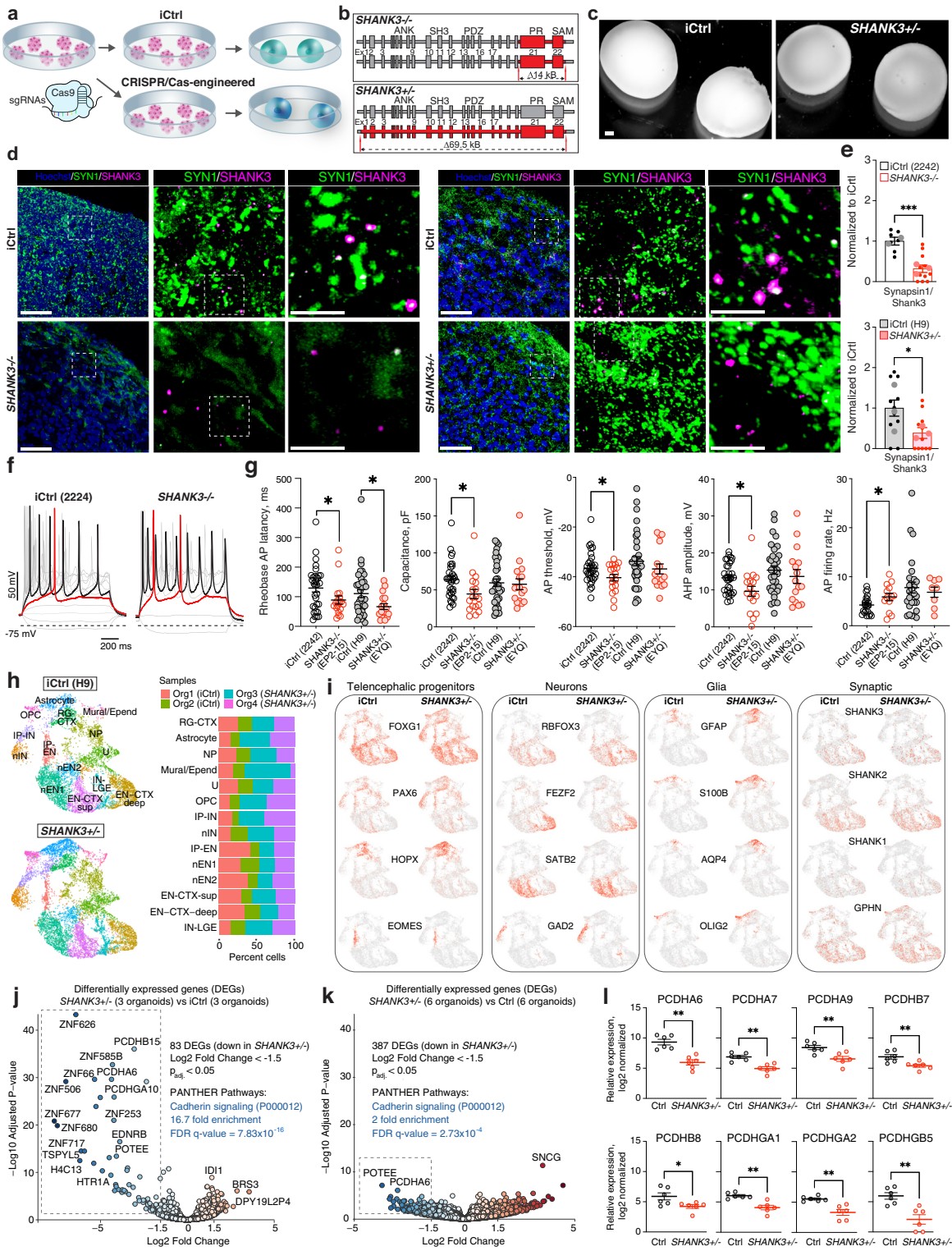

molecular pathways associated with the specification of LGE-derived INs, and identifies the genes and molecular pathways disrupted by a patient-related hemizygous *SHANK3* deletion in human telencephalic tissue. Additional studies will be needed to understand how the batch-to-batch, line-to-line, and individual-to-individual differences influence the reproducibility of germinal zone organization, size and shape of the lumen and cortical layers, and specification of different telencephalic cell types in organoids produced from single and multiple rosettes.

## Methods

The research described in this study complies with all relevant ethical regulations. It was performed in accordance to the protocols approved by the University of Utah Environmental Health and Safety committee and Institutional Animal Care and Use committee.

### Human subjects

All human subjects donated cells for this study signed an informed consent approved the Stanford University Institutional Review Board

**Fig. 10 | Cellular and molecular deficits in SHANK3-deficient organoids.**
**a, b** Generation of CRISPR/Cas9-engineered PSC lines and organoids with homozygous and complete hemizygous SHANK3 deletions (*SHANK3-/-* and *SHANK3*+/−, respectively) (elements were drawn by Coni Hoerndli Science Design). **c** Images of 5-month-old iCtrl and *SHANK3*+/− SNR-derived organoids. **d** Synaptic stainings using anti-SYN1 and anti-SHANK3 antibodies in iCtrl and SHANK3-deficient organoids (left: iCtrl [2242-5] and *SHANK3-/-* [EP2-15]; right: iCtrl [H9] and *SHANK3*+/− [EYQ]). **e** Quantification of SYN1/SHANK3 puncta (top: *n* = 7 and 12 sections from 2 iCtrl [2242-5] and 3 *SHANK3−/−* [EP2-15] organoids produced in two batches, respectively, ***P* = 0.0007, unpaired two-sided Mann−Whitney test; bottom: *n* = 11 and 11 sections from 3 iCtrl [H9] and 3 *SHANK3*+/−[EYQ] organoids produced in two batches, respectively, **P* = 0.029, unpaired two-sided Mann−Whitney test). **f** Traces of voltage deflections recorded from iCtrl (2242-5) and *SHANK3-/-* (EP2-15) neurons in response to different somatic current injections ($\Delta I$ = 10pA). **g** Quantified Rheobase AP latency, membrane capacitance, AP threshold, AHP, and AP firing rate for iCtrl and SHANK3-deficient neurons in 5-month-old SNR-derived organoid slices (*n* = 32, 18, 36, and 17 cells, from 4 iCtrl [2242-5], 3 *SHANK3-/-* [EP2-15], 11 iCtrl [H9],

and 2 *SHANK3*+/− [EYQ] organoids, [P = 0.053 and 0.039 [Rheobase AP latency], 0.012 [membrane capacitance], 0.038 [AP threshold], 0.011 [AHP], 0.041 [AP firing rate], unpaired two-sided *t*-test). These cells are also included in analyses presented in Fig. 8i and 9d. **h** UMAP visualizations of unsupervised clustering of scRNA-seq data from 2 iCtrl [H9] and 2 *SHANK3*+/− [EYQ] SNR-derived organoids and relative cluster composition (right). These organoids are also included in the clustering results presented in Fig. 5. **i** Heatmap of expression of different cell-type specific marker genes in iCtrl (H9) and *SHANK3*+/− (EYQ) organoids. **j, k** Volcano plot visualizations of DEGs from bulk RNA-seq data obtained from 3 iCtrl (H9) and 3 *SHANK3*+/− (EYQ) organoids (**j**) and 6 control and 6 *SHANK3*+/− organoids. Gene ontology (GO) analysis performed on most downregulated genes (Log2 fold change < −1.5) detected in *SHANK3*+/− organoids. **l** Relative expression of clustered protocadherins in 6 control and 6 *SHANK3*+/− organoids (P = 0.0022 [PCDHA6], 0.0022 [PCDHA7], 0.009 [PCDHA9], 0.009 [PCDHB7], 0.041 [PCDHB8], 0.0022 [PCDHGA1], 0.0022 [PCDHGA2], and 0.0043 [PCDHGB5], unpaired two-sided Mann−Whitney test). Data are mean ± s.e.m. Scale bars = 450 (**c**), and 50, 20, and 5 μm (**d**). Source data are provided as a Source Data file.

---

(IRB) committee. Deidentified human iPSC cells were transferred for experiments at the University of Utah under an MTA agreement.

### Stem cell culture and neural differentiation

Human pluripotent stem cell lines used in this study (control lines: H9 [WiCell], 2242-5 [Dolmetsch lab, Stanford University], and GM07492 [Ernst lab, McGill University (Bell et al.)[98]]; SHANK3-deficient lines: EYQ2-20 [*SHANK3*+/−, Shcheglovitov lab, (Chiola et al.)[64]], EP2-15 [*SHANK3-/-*, Shcheglovitov lab], 7349-3 [patient *SHANK3*+/−, Shcheglovitov lab]) were maintained on Matrigel (1%, BD) in Essential 8 Medium (Thermo Fisher Scientific). For neural differentiation, cells were passaged at high density to reach ~90% confluency 2−3 days post-passaging. For induction, E8 medium was replaced with neural differentiation (ND) medium containing a 1:1 mixture of N2 medium (DMEM/F-12, 1% N2 Supplement [Cat#17502048], 1% MEM-NEAA, 2 μg/ml Heparin, and 1% Pen/Strep) and B-27 medium (Neurobasal-A [Cat#10888022], 2% B27 Supplement with vitamin A [Cat#17504044], 1% GlutaMAX and 1% Pen/Strep) supplemented with SMAD inhibitors (4 μM dorsomorphin and 10 μM SB431542). ND medium was replaced daily for the next 7−10 days. On days 7−10, cells were manually scraped from the plate using a sterile glass hook. Cell clumps were re-plated on Matrigel-coated plates in ND medium with EGF (10 ng/mL) and FGF (10 ng/mL). Approximately 50−100% of media was replenished daily for the next 3−6 days to remove dying cells. Neural rosette clusters appeared after ~6 days in EGF/FGF-containing media and were manually split using a sterile glass hook and re-plated on Matrigel-coated plates at low density. On days 15−19, SNRs were manually isolated and transferred to ultra-low attachment plates in ND medium with EGF (10 ng/mL) and FGF (10 ng/mL). Plates were kept on an orbital shaker (45 rpm) in a 37 °C, 5% CO₂ incubator with 50% media exchange every other day to maintain the final concentrations of EGF and FGF at 10 ng/mL. At 2−3 weeks post SNR isolation, SNR-derived organoids with clearly visible single lumen were transferred into Matrigel as previously described (Lancaster and Knoblich, 2014) and cultured in uncoated 10-cm Petri dishes with ND medium without growth factors. The dishes with organoids were transferred to the orbital shaker 24 h post-Matrigel embedding. Approximately 50% of the media was exchanged every 3−4 days thereafter. At 3 months post SNR isolation, BDNF (10 ng/mL), GDNF (10 ng/mL), and NT-3 (10 ng/mL) were added to the culture with each media change to promote synapse development and maturation.

### Generation of isogenic stem cell line with homozygous *SHANK3* deletion

A *SHANK3*-deficient stem cell line (EP2-15) was generated using a CRISPR/Cas9 approach from the 2242-5 iPSC line. Two sgRNA sequences, gRNA #8 (TGAGCCGCTATGACGCTTCAGGG) and gRNA #4

(TTTCTCAGGGGTCCCCGGTGGGG), designed to flank exons 21-22 and the stop codon for creating an ~14-kB deletion, were introduced using electroporation together with Cas9 sequence under the control of the EF-lα promoter. Upon electroporation of 5 μM CRISPR/Cas9 plasmids into ~3 × 10⁶ cells using the Amaxa Nucleofector System (program A023), iPSCs were resuspended in E8 medium with ROCK inhibitor. The following day, cells were exposed to 0.25 μg/mL puromycin for 48 h. Antibiotic-resistant colonies were picked and expanded on Matrigel-coated plates in E8 medium. A total of 82 single colonies have been manually picked, propagated, and screened for the deletion using conventional PCR and a set of one forward and two reverse primers (Primer F [yw3]: CTCAGGGCCTGCTTGATGAC; Primer R [yw4]: GTGCGCTCCTGAAGGACAAT; and Primer R [yw7]: GGTCTTGCATC GAGGTGCTC), which recognize different regions in the proximity of exons 21-22. Out of the 82 colonies, one was identified with homozygous *SHANK3* deletion encompassing exons 21-22. The deletion was verified using PCR product purification, subcloning into a TOPO TA cloning vector (Thermo Fisher), and sequencing.

### Single-cell RNA-sequencing

**Single-cell collection from 1-month-old organoids.** SNR-derived organoids were rinsed with sterile Dulbecco's phosphate-buffered saline (DPBS, 1X) containing 5% DNaseI and then dissociated in a pre-warmed working solution of either Accutase (1 month) with 5% DNase I. The samples were incubated at 37 °C for 5 min. After removal of the enzyme supernatant, cells were resuspended in Trypsin Inhibitor with 5% DNase I, incubated for 5 min at 37 °C. Cells were washed with Neurobasal-A Medium with 5% DNaseI and briefly centrifuged before the next step. Cells were resuspended in 200 μL Neurobasal-A and manually triturated by pipetting up and down ~20−30 times. The suspension was centrifuged at 200 × *g* for 4 min and resuspended in 200−500 μL Neurobasal-A. The suspension was passed through a 40-μm cell strainer to yield a uniform single-cell suspension. Samples were kept on ice from this point forward. Samples were centrifuged at 200 × *g* for 6−8 min. After removal of the supernatant, cells were resuspended in 35−50 μL Hank's Balanced Salt Solution (1X) or DPBS (1X), and cell viability was assessed using Trypan blue staining. Samples were quickly transferred to the High Throughput Genomics Core Facility at the University of Utah.

**Single-cell collection from 5-months-old organoids.** SNR-derived organoids were subjected to vibratome slicing in ice-cold "cutting" artificial cerebrospinal fluid (ACSF), containing: KCl 2.5 mM, MgCl₂ 7 mM, NaH₂PO₄ 1.25 mM, Choline-Cl 105 mM, NaHCO₃ 25 mM, D-Glucose 25 mM, Na⁺-Pyruvate 3 mM, Na⁺-L-Ascorbate 11 mM, CaCl₂ 0.5 mM. Collected tissue slices (350 μm thick) were enzymatically dissociated using prewarmed Papain-containing solution for 20−30 min

at 37 °C. After removing enzyme supernatant, cells were resuspended in Trypsin Inhibitor with 5% DNase I, incubated for 10 min at 37 °C, and gently triturated with p1000, p200, and p20 in the trypsin inhibitor solution. The cell suspension was passed through a 40 μm cell strainer. Samples were centrifuged at 200 × g for 7 min to remove the supernatant. After removing the supernatant, cells were resuspended in ice-cold sterile PBS (1×) with 0.04% bovine serum albumin (BSA) to 1000 cells/μL. Samples were transferred on ice to the High Throughput Genomics Core Facility at the University of Utah to assess cell viability using Trypan Blue staining and to load the cells in the 10X Genomics Chromium Single Cell 3′ Microfluidics Chip (10,000 cell/channel). Using this approach, we achieved more than 90% cell viability.

**Single-cell capture and library preparation.** Single-cell capture, lysis, and cDNA synthesis were performed with the 10x Genomics Chromium at the High Throughput Genomics Core Facility at the University of Utah. All cells were processed according to 10X Genomics Chromium Single Cell 3′ Reagent Guidelines. Sequencing was performed using Illumina HiSeq 26 × 100 cycle paired-end sequencing.

**Single-cell RNA-seq processing.** Reads were processed using the 10X Genomics Cell Ranger 2.0 pipeline as previously described (Zheng et al.[99]) and aligned to the human GRCh38 reference assembly (Ensembl v84). Gene-barcode matrices were constructed for each sample by counting unique molecular identifiers (UMIs). The single-cell RNA-seq data reported in this paper is available in the GEO repository with accession number GSE211878.

### Single-cell RNA-seq analysis of 1-month-old organoids
**Data filtering.** To exclude low-quality cell libraries, we performed quality control using the scater R package (McCarthy et al.[100]) and filtered cells based on the distribution of library size and mitochondrial RNA. Cells with <2000 UMIs or 1200 detected genes and those with more than 14000 UMIs or 4000 detected genes were removed. To remove cells that contain abnormally high or low number of UMIs relative to its number of detected genes, we fit a smooth curve (loess, span = 0.5) to the number of UMIs vs. the number of genes after log-transform, and removed cells with residuals more than three standard deviations from the mean. We also removed cells with more than 6% of UMIs assigned to mitochondrial genes. After cell filtering, genes detected with at least one UMI in at least two cells were kept in the dataset.

In addition, we used the DoubletDetection (Version v2.4) Python package (https://zenodo.org/record/2678042) to remove putative doublet or multiplet cell libraries from the dataset. We run the program for 50 iterations for each sample separately, we used a conservative threshold and classified cells with $p < 10^{-5}$ in 90% of iterations as multiplets. Using this method, 2.5% of cells were removed, which agrees with the expected multiplet rate of the 10X Genomics platform (2.3% for ~3000 cells recovered). The filtered dataset containing 9577 cells were used for downstream analysis.

We performed library size normalization, feature selection, dimensionality reduction and clustering using the Seurat v2 R package (Butler et al.[101]).

**Data normalization.** The gene expression of each cell was normalized by its total UMI count with a default scale factor of 10,000, and log-transformed the normalized count for downstream analysis.

**Feature selection and scaling.** We used the FindVariableGenes function in Seurat v2 to identify a set of highly variable genes (HVGs), which are informative genes that exhibit high variance across the dataset. We reason that the gene expression values in each single cell library are determined by the combination of four variables: (1) biological variables related to cell type identity, (2) biological variables related to

common cellular activities rather than a specific cell type (e.g. cell cycle), (3) technical variables related to library size (e.g. number of UMIs), and (4) random noise. To focus our downstream analysis on gene expression programs involved in cell type identity, we performed careful selection of HVGs to eliminate the effects unrelated to cell type. To exclude genes with very low expression and housekeeping genes from HVGs, we only considered genes that are detected in at least 10 cells but <90% of all cells, with average expression between 0.03 and 3. Among these genes, the top 1200 genes ranked by scaled dispersion were selected as HVGs. We further excluded ribosome and mitochondrial gene (as identified by gene names starting with RPL, RPS or MT) from HVGs.

To remove the effect of cell cycle signals on dimensionality reduction and cell type clustering, we used the list of 97 cell cycle genes provide by the Seurat v2 package as the set of known cell cycle genes. Using this information, we first removed 53 known cell cycle genes present in the initial set of HVGs. Then we used these genes as reference to identify other putative cell cycle genes in HVGs by correlation analysis: (1) we collected a subset of robust cell cycle genes that show strong correlation (spearman correlation > 0.3) with any other known cell cycle genes; (2) next, we identified another 88 genes in HVGs that are highly correlated (spearman correlation > 0.3) with any cell cycle gene in the above subset. These putative cell cycle genes were subsequently removed from HVGs. The final set of HVGs consisted of 1056 genes.

To remove the sample-specific batch effect, we performed batch correction using the mutual nearest-neighbors correction method in the scran R package (Haghverdi et al.[102]). Batch correction was performed on HVGs with the sigma value of 0.5.

The batch-corrected HVG expression matrix was then fed into the ScaleData function in Seurat v2. The ScaleData function removes the effect of library size on cell-cell variation in gene expression by fitting a linear regression model on each gene using the number of detected UMIs as a predictor. The residuals were then scaled to zero-mean and unit-variance.

**Dimensionality reduction.** To reduce the dimensionality of the dataset, we performed principal component analysis (PCA) on the scaled HVG expression matrix. The first 21 principal components were chosen for downstream analysis, as judged by the PCA elbow plot and the heatmap distribution of top-ranked genes for each principal component.

**Cell type clustering.** Clustering of single cells were performed using the FindClusters function in Seurat v2. Briefly, we built a shared nearest neighbor graph of single cells using the first 21 principal components and a neighborhood size of 30. Louvain clustering with resolution of 1.2 was then performed on the single cell graph to partition cells into discrete clusters. We computed a cosine similarity matrix between all pairs of clusters using cluster-averaged expression profile and constructed a cluster dendrogram using (1 - cosine_similarity) as the distance metric and the average linkage method.

**Differential expression.** To find marker genes that are enriched in each cluster, we performed differential expression analysis between each cluster and the rest of the dataset using the Wilcoxon test in the Seurat v2 FindAllMarkers function. We require marker genes to be present in at least 25% of cells in the cluster, have a fold change of 2 by average expression level, a fold change of 2 by the percentage of expressing cells, and a Bonferroni-corrected $p$ value < 0.01.

**Nonlinear dimensionality reduction.** To embed single cells into a two-dimensional visualization, we took the above principal components as inputs and performed nonlinear dimensionality reduction using the Uniform Manifold Approximation and Projection (UMAP) algorithm

(McInnes and Healy[103]) with n_neighbors = 20 and min_dist = 0.3. The resulting UMAP embedding attempts to preserve both the local neighbor relations and the global structure of the data.

**RNA velocity analysis.** RNA velocity analysis was performed using the Velocyto Python package (La Manno et al.[48]) to infer the future states of single cells during development. Briefly, RNA velocity vectors for each cell were calculated using the relative abundance of spliced and unspliced mRNAs of each gene. Spliced and unpliced read counts were generated using the Velocyto command line interface. Genes with low spliced counts (<40 UMIs or detected in <30 cells) were excluded from analysis. The top 3000 variable genes were selected by fitting the variance vs. mean relationship.1648 variable genes are kept after filtering by unspliced counts (<25 UMIs or detected in <20 cells), cluster-averaged expression (unspliced < 0.01 or spliced < 0.08 in all clusters), as well as removing ribosome, mitochondria and cell cycle genes. PCA was performed on variable genes and the first 20 principal components were used to build a k-nearest neighbor graph (KNN) with k = 200. The velocity vector was inferred after KNN imputation and the future state of each cell was extrapolated using the constant velocity model and a time step of 1. To project the velocity vectors onto the single cell UMAP embedding, the transition probabilities between pairs of cells were estimated with a neighborhood size of 750 and the embedding shift was calculated with sigma = 0.05. The smoothed velocity vector grid was calculated with a smoothing factor of 0.8 and neighborhood size of 300. To simulate the progression of single cells through developmental time, Markov Chain simulation was performed for 200-time steps.

**Smoothed gradient of gene expression change using Slingshot.** As UMAP constructs a low-dimensional embedding of the data manifold that captures both local and global structure of the data, we fit smooth curves along the UMAP embedding to study how gene expression changes across cell types. The Slingshot algorithm (Street et al.[104]) was used to fit both linear and branching curves across the single cell clusters. To constrain the smooth curve fitting algorithms to recover biologically meaningful solutions, we (1) performed the fit separately on the progenitor clusters, the excitatory lineage and the inhibitory lineage, and (2) used the Markov Chain endpoint clusters from RNA velocity inference as the supervised end points in the Slingshot algorithm. This method recovered smooth curve trajectories that agrees well with the RNA velocity analysis. Cells along each lineage were projected onto the Slingshot trajectory curves and ordered according to the assigned pseudotime values. Generalized additive model from the gam R package was used to identify genes that are differentially expressed along each trajectory lineage. Genes with small fold change (<6 for neural differentiation trajectories or <3 for the progenitor trajectories) were removed from the list of differentially expressed genes. Genes were assigned to early- or late-stage by the time of their peak expression along the trajectory. A set of common early EN/IN differentiation genes were produced by the intersection of early EN/IN genes across neural differentiation trajectories.

**Gene Ontology (GO) analysis.** GO analysis was performed using the ToppGene Suite (Chen et al.[105]).

**Jaccard similarity analysis.** The top 100 differentially expressed genes for fetal human CTX, LGE, and MGE at three different developmental stages, PCW 7, 9, and 11, were selected by ranking the region-specific genes identified by from Bocchi et al.[8] using the lowest *p*-adjusted values for each comparison per area (e.g. CTX vs LGE and CTX vs MGE), summing the p-adjusted values of each comparison, and then taking the top 100 genes with the lowest summed p-adjusted values (Supplementary Data 4). For the organoids, we ranked the top 1000 genes per cell cluster using the Wilcoxon rank-sum test

(Supplementary Data 4). Then, the two lists were compared to assess their similarity using the Jaccard index.

**Single-cell RNA-seq analysis of 5-month-old organoids**
Single-cell RNA-seq analysis of the six 5-month-old organoids was performed using the similar methods as the 1-month-old organoids.

**Data filtering and normalization.** To exclude low-quality cell libraries, we filtered cells based on the distribution of library size and mitochondrial RNA. In Org5, cells with <2000 UMIs or 750 detected genes and those with more than 23,000 UMIs or 3800 detected genes were removed. For the other five organoids, cells with <1300-1500 UMIs or 700 detected genes and those with more than 35,000 UMIs or 7000 detected genes were removed. To remove cells that contain abnormally high or low number of UMIs relative to its number of detected genes, we fit a smooth curve (loess, span = 0.5) to the number of UMIs vs. the number of genes after log-transform, and removed cells with residuals more than three standard deviations from the mean. We removed cells with more than 6-10% of UMIs assigned to mitochondrial genes. After cell filtering, genes detected with at least one UMI in at least two cells were kept in the dataset. We performed multiplet detection using DoubletFinder (McGinnis et al., 2019) in organoid samples with more than 1500 cells and removed the putative multiplets. The preprocessed individual datasets contain 4682, 2823, 4961, 4618, 1152, 4250 cells, respectively.

**Feature selection and scaling.** We selected 3000 HVGs separately for each of the six 5-month-old organoids using the SCTransform function in Seurat v3 (Hafemeister and Satija[106]; Stuart et al.[107]). SCTransform also normalized and scaled the expression matrix. Cell cycle genes were then removed from HVGs following the same approach as in the 1-month-old organoids.

**Integration of the 5-month-old organoid datasets.** To perform integrated analysis on the six 5-month-old datasets, we combined them together using the single-cell integration pipeline in Seurat v3. The set of gene features used is the union of HVGs from all datasets. To ensure all features are present in each dataset, we padded zeros rows when a gene within the union features is absent in a dataset's gene expression matrix. We selected 3000 integration features using the SelectIntegrationFeatures function, and prepared the dataset for integration using the PrepSCTIntegration function. A set of "anchors" were identified between pairs of datasets using the FindIntegrationAnchors function (dims = 1:30, k.anchor = 3, k.filter = 20, k.score = 15). Then, we integrated all datasets together using these anchors and the IntegrateData function (dims = 1:30, k.weight = 50). Clustering was performed on the integrated dataset using a resolution of 0.55, a neighborhood size of 20 and the first 30 principle components. UMAP was performed with n_neighbors = 20 and min_dist = 0.3. Subsequent differential expression analysis and marker gene selection was done using the same method as in the 1-month-old organoids. To analyze the correlation of cell type clusters between a pair of different datasets, we computed the cluster-averaged gene expression profile for each cluster using the intersection of HVGs from the two datasets and the scaled expression values. Pearson correlation was then performed between clusters from the two datasets.

**Bulk RNA-seq and correlation analysis**
Samples from 5-month-old SNR-derived organoids were homogenized in Trizol (Invitrogen), and total RNA was purified using the RNeasy Micro Kit (Qiagen) with DNase I digestion. Ribosomal RNA was depleted with the NEBNext rRNA Depletion Kit. Libraries were prepared using the NEBNext Ultra Directional RNA Library Prep Kit for Illumina and verified using the Bioanalyzer RNA Nano Assay (Agilent) before sequencing on the NextSeq 500 (Illumina). Single-end reads

were trimmed to 70 bp for genomic alignment. Total RNA-seq reads for SNR-derived organoids and fetal human brain samples were aligned to hg38 (i.e., GRCh38.p2) using BWA and further processed for exonic expression levels (RPKM) across ~24,000 RefSeq-annotated genes using a previously described pipeline[55]. Reads that did not map uniquely with at most two mismatches were excluded.

The correlation analysis was performed on data from seven SNR-derived organoids and 10 fetal-brain samples[55], plus 15 samples from four published studies available through the NCBI Gene Expression Omnibus (GEO) site: GSM2580319, GSM2580321, GSM2580323, GSM2580325, GSM2580327, and GSM2580329 (ref. 18); GSM2112671 and GSM2112672 (ref. 17); GSM3408648, GSM3408667, and GSM3408685 (ref. 54); and GSM2180144, GSM2180145, GSM2180138, and GSM2180139 (ref. 22). Twelve datasets of FASTQ reads (GSM2580319, GSM2580321, GSM2580323, GSM2580325, GSM2580327, GSM2580329, GSM2112671, GSM2112672, GSM2180144, GSM2180145, GSM2180138, and GSM2180139) were run through the same pipeline to obtain expression levels over the same gene set. The other three datasets (GSM3408648, GSM3408667, and GSM3408685), only available as read counts per gene, were normalized to each gene's total exonic length in order to be comparable to an RPKM-like expression density (to within an overall sample-dependent scaling factor). The data for each pair from among these 32 datasets were filtered to exclude snoRNA genes (some of which were expressed at high levels in a highly variable manner from sample to sample) and any genes that had precisely zero expression in either sample. This typically yielded ~15,000–19,000 informative pairs of expressed genes per sample pair, for which Spearman correlations were calculated.

The BrainSpan dataset was obtained from http://www.brainspan.org/. These data altogether included 578 samples from 41 different donors spread across 30 ages ranging from 8 pcw through 40 years old and covering 26 brain regions. Expression levels (RPKM) in each BrainSpan sample were provided for 18,529 genes found in the RefSeq annotation for hg19 (GRCh37). The regions (a) ventrolateral prefrontal cortex (VFC), (b) medial prefrontal cortex (MFC), and (c) dorsolateral prefrontal cortex (DFC) included, respectively, 38, 37, and 39 BrainSpan samples at any age. The genome-wide expression profiles of the seven SNR-derived organoid samples in Supplementary Fig. 13 were analyzed with respect to hg38; for direct comparison to the BrainSpan samples in these regions, 19,889 hg38 genes were directly translatable to the hg19 RefSeq annotation, of which 18,448 genes overlapped genes documented by BrainSpan. A Spearman correlation coefficient was then calculated between each organoid sample and each of BrainSpan sample for each brain region, after removing genes with zero expression in either sample pair.

To identify the differentially expressed genes and pathways in control and SHANK3+/− organoids (Fig. 10), the human GRCh38 genome and gene annotation files were downloaded from Ensembl release 102 and a reference database was created using STAR version 2.7.6a with splice junctions optimized for 150 base pair reads. Optical duplicates were removed from the paired end FASTQ files using clumpify v38.34 and reads were trimmed of adapters using cutadapt 1.16. The trimmed reads were aligned to the reference database using STAR in two pass mode to output a BAM file sorted by coordinates. Mapped reads were assigned to annotated genes using featureCounts version 1.6.3. The output files from cutadapt, FastQC, FastQ Screen, Picard CollectRnaSeqMetrics, STAR, and featureCounts were summarized using MultiQC to check for any sample outliers[108]. Differentially expressed genes were identified using a 5% false discovery rate with DESeq2 version 1.30.1[109] and pathways were identified using PANTHER[110].

## Electrophysiology

**Slice preparation.** Acute slices for electrophysiology experiments were obtained from 5-month-old SNR-derived organoids. SNR-derived organoids were placed in ice-cold "cutting" artificial cerebrospinal fluid (ACSF), containing: KCl 2.5 mM, MgCl$_2$ 7 mM, NaH$_2$PO$_4$ 1.25 mM, choline-Cl 105 mM, NaHCO$_3$ 25 mM, D-glucose 25 mM, Na$^+$-pyruvate 3 mM, Na$^+$-L-ascorbate 11 mM, and CaCl$_2$ 0.5 mM. Tissue sections (200-μm-thick) were cut with a Leica VT1200 vibratome in ice-cold "cutting" ACSF. Sections were allowed to recover for 30 min in standard ACSF in a 37 °C water bath before being stored at room temperature in standard ASCF. Standard ACSF contains: NaCl 124 mM, KCl 2.5 mM, MgCl$_2$ 1 mM, NaH$_2$PO$_4$ 1.25 mM, NaHCO$_3$ 25 mM, D-glucose 25 mM, Na$^+$-pyruvate 1 mM, Na$^+$-L-ascorbate 1 mM, and CaCl$_2$ 2 mM. Both "cutting" and standard ACSF were bubbled with 95% O$_2$/5% CO$_2$.

**Patch-clamp recordings.** Individual slices were transferred to a recording chamber and submerged in standard ACSF bubbled with 95% O$_2$/5% CO$_2$. To characterize the electrophysiological properties of individual cells, whole-cell patch-clamp recording was performed at room temperature under constant perfusion of standard ACSF. Cells were visualized under a 40x water-immersion objective mounted on an upright microscope equipped with infrared differential interference contrast and a digital camera. Whole-cell patch-clamp recordings were performed using glass pipette electrodes (resistance 3–7 MΩ) pulled from borosilicate capillaries (BF150-86-10, Sutter Instruments, Novato, CA, USA) using a P-97 pipette puller (Sutter Instruments). Recording data were acquired using a Multiclamp 700B amplifier (Molecular Devices, Palo Alto, CA, USA) with Bessel filter of 2 kHz and digitized at 50 kHz using Axon DigiData 1500A (Molecular Devices). The Axon pClamp 10 (Molecular Devices) software was used to control the experiment. Neurons residing in the layered regions of organoid slices and demonstrating pyramidal-like morphology were selected for the experiments.

To characterize the intrinsic electrical properties of neurons, membrane voltage was recorded in the current clamp configuration and electrodes were filled with intracellular solution containing KMeS 123 mM, KCl 5 mM, HEPES 10 mM, Na-ascorbate 3 mM, MgCl$_2$ 4 mM, Na$_2$-phosphocreatine 10 mM, Na$_2$GTP 0.4 mM, and Na$_2$ATP 4 mM. 0.5% biocytin was added for a subset of experiments to visualize neuronal morphology. The resting membrane potential was recorded in the I = 0 mode within 30 s after establishing the whole-cell patch. The measurements of the resting membrane potential were corrected for the liquid junction potential (9.4 mV). A small bias current was applied to maintain the baseline membrane voltage between −60 ~ −75 mV. Square-pulse currents (1 or 2 s) were injected every 5 s, starting from hyperpolarizing amplitudes to increasingly more depolarized amplitudes, with a step size of 5 or 10 pA determined by the input resistance of the cell. For consistency, only the first 1 s of the current clamp data was analyzed. After patch-clamp recordings, sections were fixed in 4% paraformaldehyde (PFA) in PBS at 4 °C overnight for subsequent immunohistochemical staining.

**Current-clamp data analysis.** To automate the analysis of current clamp data, we built a data analysis pipeline in Python using the DataJoint relational database[111] and custom-written scripts adapted from the AllenSDK package (Allen SDK, 2015). The Axon binary files were converted into Python arrays using the stfio package from Stimfit[112]. We used the ephys module from AllenSDK as a baseline analysis framework and added methods to extract eletrophysiological features from our current clamp data. The analysis code was called within the DataJoint data processing pipeline to automatically analyze all recording data. The analysis results and experimental metadata were stored in the DataJoint relational database. Below we detail the analysis of each electrophysiological feature.

Rheobase sweep was determined as the first depolarizing sweep that has at least one action potential in the first 500 ms. Single action potential properties were measured using the first action potential of

the rheobase sweep. Spike threshold was defined as the voltage where $\Delta V/\Delta t$ achieved 5% of the upstroke (maximum rising phase $\Delta V/\Delta t$). Spike amplitude was measured as the difference between the peak voltage and the threshold voltage. Spike width was measured at the half-height of the spike. Spike trough (fast) was measured at five times spike width after the spike threshold. After-hyperpolarization (AHP) was defined as the difference between the trough voltage and the threshold voltage.

First-spike latency was measured at 5 pA above the rheobase current. Since not all recordings have a current injection step size of 5 pA, the first-spike latency at rheobase + 5 pA was linearly interpolated using the two sweeps right above and below the target current.

The adaptation index was calculated on a subset of recordings that had at least three supra-threshold sweeps each with at least four spikes. Cells that did not meet these criteria had an undefined adaptation index. For each pair of consecutive inter-spike intervals (ISI), the adaption index was calculated as $(ISI_2 - ISI_1) / (ISI_1 + ISI_2)$. Since different sweeps and recordings may have different numbers of spikes, we calculated the averaged adaptation index for each sweep using only the first two adaptation indices. We then averaged the adaptation indices of the first three supra-threshold sweeps as the adaptation index of each cell.

The F-I curve slope was calculated by a linear fit of firing rate against current injection amplitude. Data points after the maximum firing rate was reached were excluded from the fit. Cells with a maximum firing of only one spike per sweep were assigned a F-I slope of zero.

Input resistance was calculated using a linear fit of the maximum voltage deflection of each hyperpolarizing sweep against the current injection amplitude. The membrane time constant (tau) for each hyperpolarizing sweep was calculated by fitting a single exponential from 10 to 100% of the maximum voltage deflection. Membrane capacitance was then calculated by first averaging the time constant of all hyperpolarizing sweeps and then dividing by input resistance.

The sag ratio for each hyperpolarizing sweep was calculated as $(V_{peak} - V_{steady\text{-}state}) / (V_{peak} - V_{baseline})$, where $V_{peak}$ is the maximum voltage deflection within the first 500 ms and $V_{steady\text{-}state}$ is the averaged membrane voltage during the last 100 ms of the 1-s current injection. For each cell, two sag ratios with $V_{peak}$ closest to −100 mV were averaged.

**Hierarchical clustering.** Using the above electrophysiological features, we constructed a cell-feature matrix. Hierarchical clustering was performed using the average linkage method and correlation distance metric. Briefly, each feature column was scaled to zero-mean and unit-variance, and the correlation distance was calculated on scaled features. For each pair of cells, if either of them had undefined adaptation index, the correlation was calculated without using adaptation index. Some features (spike width, first-spike latency, input resistance, and capacitance) had skewed distribution as judged by quantile-quantile plot and were log-transformed for distance calculation. After hierarchical clustering, the dendrogram tree was cut and 5 clusters was used for downstream analysis.

**Synaptic current recordings.** To measure spontaneous synaptic transmission from neurons in organoid slices, whole-cell voltage clamp was performed. The recording electrode was filled with intracellular solution containing CsMeS 120 mM, CsCl 5 mM, HEPES 10 mM, CaCl2 0.5 mM, EGTA 2 mM, Na-Ascorbate 3 mM, MgCl2 4 mM, Glucose 5 mM, Na2-phosphocreatine10 mM, Na2GTP 0.4 mM, Na2ATP 4 mM. The membrane voltage was held at −70 mV for recording spontaneous glutamate-mediated excitatory post-synaptic currents and +10 mV for recording spontaneous GABA-mediated inhibitory post-synaptic

currents. The synaptic currents were analyzed using the Clampfit 10 software.

**Multi-electrode array recordings.** Five-month-old organoids were manually implanted with thin and flexible 16-electrode recording arrays (MicroFlex, Blackrock Microsystems). The electrodes on the array are spaced 200 μm apart, spanning 3.2 mm length. The connector of each array was secured to the side of a 60 mm Petri dish to enable connection to a stimulation/recording headstage (RHS2116, Intan Technologies) without dish opening. The headstage interfaced the RHS2000 Stimulation/Recording Controller (Intan Technologies). Data were collected from 6-month-old organoids, one-month post-implantation. For recordings, a dish with an organoid was transferred onto a heated platform to maintain 37 °C during the experiment. The protocol for data acquisition consisted of 5-min baseline recording of spontaneous activity that followed by 5-min segments of electrical microstimulation of the most active channel using single 100 μs long biphasic pulses separated by 60 s interval or bursts of 5-pulse at 100 Hz. The currents of 1, 10, and 100 μA were used for stimulation during different 5-min segments.

The recorded signals were low-pass filtered at 7 kHz and sampled at 30 kHz to avoid aliasing. The data analyses were performed using custom MATLAB scripts. In particular, the frequency analysis of Fig. 9o was performed using the spectrogram function in MATLAB over 0.2 s signal windows. To obtain estimates of discharge rates (Fig. 9p), low-frequency components were filtered out with a 300 Hz high-pass filter, and the resulting signals thresholded at 40 μV. The number of waveforms exceeding that threshold per second yielded the rate estimate.

### Western blotting

Protein lysates were prepared with Laemmli Buffer (Bio-Rad, 1610737) and loaded onto a sodium dodecyl sulfate (SDS)-polyacrylamide gel electrophoresis (PAGE) chamber to separate protein samples. Separated proteins were transferred to a polyvinylidene difluoride (PVDF) membrane overnight at 4 °C. Following transfer, membranes were blocked with 5% milk in PBS with 0.1% Tween 20 (PBST) for 1–2 h while rocking at room temperature. Membranes were then incubated in primary anti-SHANK3 (Santa Cruz Biotechnology, sc-30193) at 1:350, or anti-TUJ1 (Covance, MMS-435P) at 1:38,000 in 5% milk in PBST and slowly rocked at 4 °C overnight. Membranes were washed three times for 10 min with PBST and then incubated with horseradish peroxidase (HRP)-conjugated secondary antibody (HRP anti-rabbit [Jackson ImmunoResearch, 111-035-144] or HRP anti-mouse [Vector Laboratories, PI-2000) at 1:15,000-1:25,000) in 5% milk in PBST for 1–2 h at room temperature. After three 10-min washes in PBST, the membrane was placed in deionized water and gently dried before adding chemi-luminescent HRP substrate (EMD Millipore, WBKLS0100) for visualizing proteins. Blotting X-ray film (Genesee Scientific, 30-507 L) was then placed over the membrane with a sheet protector in between. The film was processed using a Konica SRX-101 machine. Films were imaged with ChemiDoc XRS + System with Image Lab Software (Bio-Rad, 1708265) and analyzed using ImageJ.

### Immunohistochemistry and imaging

**SNR-derived organoids.** Rosettes were fixed in 4% paraformaldehyde for 15 min at room temperature, washed three times with PBS, followed by permeabilization with 0.3% Triton-X for 15 min at room temperature. Rosettes were then blocked in 3% BSA with or without 10% Normal Goat Serum for 1 hour at room temperature. Primary antibodies (Supplementary Table 2) were diluted in 3% BSA and applied over night at 4 °C. Following primary antibody incubation, rosettes were washed three times with PBS. Secondary antibodies (Supplementary Table 2) were diluted in 3% BSA and applied for 2 hours at room temperature in the dark. Rosettes were then washed three times with PBS and

incubated with Hoechst 33342 nuclear counterstain (1:50,000 in PBS) for 3 min at room temperature.

SNR-derived organoids were fixed in 4% paraformaldehyde overnight at 4 °C and washed three times with PBS before passing through a sequential sucrose gradient of 10%, 20%, and 30% sucrose solutions. SNR-derived organoids were then placed in cryosectioning molds, embedded in OCT and flash frozen, then stored at −80 °C. SNR-derived organoids were cryosectioned using a Leica Cryostat machine, and sections were cut at 10–20 μm thickness and adhered to positively charged microscope slides. Adhered cryosections were then processed for immunostaining as described above. All samples were mounted with coverslips using Aqua Poly Mounting Media (Polysciences, Inc.). Fluorescent imaging was performed at the University of Utah Cell Imaging Core Facility using Nikon A1, Nikon A1R, or Zeiss 880 Airyscan confocal microscopes. Data processing and quantification was performed using Fiji, Volocity 5.1, and/or CellProfiler by setting the threshold to remove background and identify cell bodies or nuclei as objects. Cell percentages were obtained by counting the number of objects staining positive for a marker as a ratio of the total number of nuclear objects identified by Hoechst staining. For quantification of MAP2-expressing cells in control and SHANK3-deficient organoids, images were analyzed by two independent investigators blinded to the genotypes. For quantification of synaptic puncta, organoids sections were imaged using Zeiss 880 Airyscan microscope using a 63x oil objective in the proximity to the external border. 3-4 random 60 μm² regions of interest (ROI) were selected for imaging by the investigators blinded to the genotypes. ROI images were collected at multiple focal planes (step = 0.5 μm) to cover 20–30 μm depth. The maximum projection images obtained using ImageJ were used for synaptic puncta identification and quantification using a custom python script. *Shank3*−/− adult mouse brain tissue, provided by Dr. Yong-Hui Jiang and Dr. Xiaoming Wang (DukeUniversity), was used as a negative control to optimize the threshold for detecting SHANK3-containing puncta.

**Primary human fetal brain tissue.** Postmortem human brain specimens were obtained from University of Cambridge, UK upon pregnancy terminations. All procedures were approved by the research ethical committees and research services division of the University of Cambridge and Addenbrooke's Hospital in Cambridge (protocol 96/85, approved by Health Research Authority, Committee East of England−Cambridge Central in 1996 and with subsequent amendments, with the latest approved November 2017). Tissue was handled in accordance with ethical guidelines and regulations for the research use of human brain tissue set forth by the National Institute of Health (NIH) (http://bioethics.od.nih.gov/humantissue.html) and the World Medical Association Declaration of Helsinki (http://www.wma.net/en/30publications/10policies/b3/index.html). Embryonic and fetal age was extrapolated based on the last menstrual period of the mother and crown to rump length (CRL) measured by the clinicians. The tissues were dissected, fixed (24 h, 4 °C) in 4% paraformaldehyde (PFA) and incubated in 30% sucrose for 24 h at 4 °C. Brains were than snap frozen in optimal cutting temperature (OCT) medium, sectioned (15 μm) using a cryostat (Leica, Germany) and stored at −80 °C. Sections were thawed just prior to staining and rinsed in PBS. Fixed tissues were exposed to antigen retrieval step with 1 mM EDTA solution for 60 min at 60 °C. After washing using 1X PBS, tissues were blocked and permeabilized using 5% normal goat serum (NGS; Vector) plus 0.1% Triton X-100 in PBS for 1 h at room temperature. After washing with PBS plus 0.1% Tween-20, cells were incubated overnight at 4 °C with primary antibodies (GSX2: rabbit 1:200, Merck-Millipore ABN162; PAX6 mouse 1:80, DSHB) diluted in solution containing 2.5% NGS and 0.02% Triton X-100. Alexa Fluor secondary antibodies (ThermoFisher scientific) were used 1:1000 in PBS for 1 h RT, followed by 3× washes with PBST

and 10 min incubation at RT with 1:10,000 of DAPI (ThermoFisher scientific) in PBS. Images were acquired with Leica confocal TCS SP5 (Leica Microsystem).

For RNA-FISH, 9 weeks human fetal brain sections were dehydrated in 5 min steps in 70% and 100% ethanol, respectively. A hydrophobic barrier was created around the sections and air dried for 10 min. Sections were incubated with RNAscope® Hydrogen Peroxide solution (ACD, 322335) for 10 min at RT. After incubation, slides were washed twice with ddH₂O. Protease plus (ACD, 322331) was added to the samples and incubated in a HybEZ oven for 30 min at 40 °C. After incubation, slides were washed in ddH₂O and RNAscope in situ hybridizations were performed according to the manufacturer's instructions, using the RNAscope® Multiplex Fluorescent Assay v2 (Advanced Cell Diagnostics) for fixed frozen tissue. Following probes were used (indicated with gene target name for human, respective channel, and catalogue number all Advanced Cell Diagnostics): Pax6 (Ch1, 588881) and Gsx2 (Ch4, 540601). Probe hybridization took place for 2 h at 40 °C and slides were then rinsed in 1 × wash buffer, followed by four amplification steps (according to protocol). Brain sections were then labeled with DAPI, and mounted with Prolong Gold mounting medium (P36930, Thermo Fisher Scientific). Slides were stored at 4 °C before image acquisition using a Nikon Eclipse Ti-E microscope (Nikon Instruments).

### Online browsers
The UBrain browser (http://organoid.chpc.utah.edu) web application contains interactive visualization of the single-cell RNA-seq and eletrophysiology data from this manuscript. We developed and open-sourced the Shiny Single Cell Browser software (https://github.com/yueqiw/shiny_cell_browser) in R to build the single-cell RNA-seq browser. The electrophysiology browser was developed in Python with the source code available in https://github.com/yueqiw/ephys_analysis.

### Statistics and reproducibility
All experiments were repeated at least three times. Cell lines used in different experiments are indicated in Supplementary Table 1. Sample sizes are reported in each figure legend. They were chosen based on the results of our previous studies with iPSC-derived neurons and the level of variability observed in the pilot experiments with organoids. Organoids were excluded from the experiments and analysis based on the following pre-established criteria: (1) the organoids failed to grow during the two weeks expansion phase in EGF/FGF; (2) the organoids showed multiple clearly visible lumens at the time of embedding in Matrigel; (3) <50% recovered cells (trypan blue negative cells) after dissociation for single-cell RNA sequencing experiments; (4) the single-cell RNA sequencing sample contained <500 recovered cells after sequencing. Cells were excluded from single-cell RNA sequencing analysis based on the following criteria: (1) cells with abnormally low or high amount of unique molecular identifier (UMI) counts, determined by the data distribution; (2) cells with more than 6-10% of UMIs assigned to mitochondrial genes, as judged by the data distribution; (3) Putative multiplets as determined by DoubletFinder or DoubletDetection software package. Organoids were randomly selected for experiments and were processed in a blinded manner when comparisons performed. The statistical test used for data analysis and comparison is indicated in each figure legend. When parametric tests were used for analysis, the normal distribution of data was confirmed using the Kolmogorov−Smirnov test. Microsoft Excel for Mac (v16.52) and GraphPad Prism 9 for Mac were used for statistical analyses.

### Reporting summary
Further information on research design is available in the Nature Research Reporting Summary linked to this article.

## Data availability

The data generated and analyzed in this study are included in this published article (and its supplementary information files) or available from the corresponding author upon reasonable request. Bulk and single-cell RNA sequencing data sets generated in this study were deposited in the Gene Expression Omnibus (GEO) of the National Center for Biotechnology Information (NCBI), under the following accession number: GSE210960. Interactive visualization of our transcriptomic and electrophysiological data is provided in our online browser (UBrain Browser: http://organoid.chpc.utah.edu). Source data are provided as a Source Data file. Source data are provided with this paper.

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

## Acknowledgements

The authors are thankful to Dr. Ricardo Dolmetsch and Dr. Joachim Halmayer for recruiting individuals and sharing human iPSC lines (supported by NIMH R33MH087898 grant to R.D. and J.H.); to Dr. Carl Ernst for sharing human iPSC line; Dr. Theo Palmer for assistance with material transfer; to Laura Bell for help with immunostaining, imaging, and image quantifications; to Dr. Jianmin Zhang for help with cryo-sectioning; to Dr. Nika Gurianova for help with mRNA extraction; to Oleksandr Stepanenko, Dmitro Zhmurko, Roman Savitskyi, David Richardson, Trevor Tanner, and Dr. Thomas Cheatham for help with web-browser; to Dr. Dmitri Yatsenko for help with establishing Data-Joint electrophysiology analysis pipelines; to Dr. Jian Zhou for discussion of single-cell RNA-seq analysis methods; to Dr. Brian Dalley, Opal Allen, Chris Stubben, Brian Lohman and to High-throughput genomics facility at the University of Utah for help with sequencing and data analysis; to Cell imaging core at the University of Utah for use of microscopes; to Dr. Colin Maguire and Stem cell facility at the Cellular Translational Research Core, University of Utah for help with stem cells; to Dr. Megan Williams, Dr. Monica Vetter, Dr. Sungjin Park, Dr. Richard Dorsky, Dr. Jan Christian, and Dr. Ingmar Riedel-Kruse for sharing reagents and/or commenting on the manuscript; to Dr. Yong-Hui Jiang and Dr. Xiaoming Wang for sharing WT and SHANK3–/– mouse brain tissue. The study was supported by the NIMH (R01 MH113670), NINDS (R01NS123849 and R21NS104963), Utah Neuroscience Initiative, and Utah Genome Project grants (to A.S.), and the NIH Developmental Biology Training Grant (T32HD007491) to (to G.Y.).

## Author contributions

Y.Wang, S.C., C.R., C.J.A., Y.Wu, J.S., P.T., H.M.A.U., N.U.E., A.N.C., E.V., D.B., J.C., J.K., A.S.—data acquisition, analyses, and interpretation, GY, D.A.H., V.D.B.—data analysis and interpretation; E.C.—materials, reagents, and supervision of the experiments on primary fetal human brain tissue; A.S.—conceived the idea, wrote the manuscript, and supervised all study aspects; all authors commented on the manuscript.

## Competing interests

The authors declare no competing interests.

## Additional information

Yueqi Wang [1,2], Simone Chiola [1], Guang Yang[1,2], Chad Russell[3], Celeste J. Armstrong [1], Yuanyuan Wu[1], Jay Spampanato[4], Paisley Tarboton [3], H. M. Arif Ullah [1], Nicolas U. Edgar[1], Amelia N. Chang[5], David A. Harmin[5], Vittoria Dickinson Bocchi[6,7], Elena Vezzoli[6,7], Dario Besusso[6,7], Jun Cui[8], Elena Cattaneo [6,7], Jan Kubanek [3] & Aleksandr Shcheglovitov [1,2,3] ✉

[1]Department of Neurobiology, University of Utah, Salt Lake City, UT, USA. [2]Neuroscience Graduate Program, University of Utah, Salt Lake City, UT, USA. [3]Department of Biomedical Engineering, University of Utah, Salt Lake City, UT, USA. [4]Department of Neurosurgery, University of Utah, Salt Lake City, UT, USA. [5]Department of Neurobiology, Harvard Medical School, Boston, MA, USA. [6]Department of Biosciences, University of Milan, Milan, Italy. [7]Istituto Nazionale di Genetica Molecolare, Milan, Italy. [8]Department of Cell Biology and Neurosciences, Montana State University, Bozeman, MT, USA. ✉e-mail: alexsh@neuro.utah.edu

