## [Peer Review File · Nature Communications]

Modeling human telencephalic development and autism-associated SHANK3 deficiency using organoids generated from single neural rosettesEditorial Note: This manuscript has been previously reviewed at another journal that is not operating a transparent peer review scheme. This document only contains reviewer comments and rebuttal letters for versions considered at *Nature Communications*.

REVIEWER COMMENTS

Reviewer #1 (Remarks to the Author):

The authors did a good job responding to my criticisms. The new protocol is very exciting and the data on shank3 adds new biology to the function of this gene. Thus, I support this publication.

Reviewer #2 (Remarks to the Author):

Modeling human telencephalic development and autism-associated SHANK3 deficiency using organoids generated from single neural rosettes

Wang and colleagues addressed most of my original concerns. The manuscript was appropriately edited to address the points that we raised. Substantial new analysis and experiments were added including comparisons with human in vivo transcriptomic data, evaluation of reproducibility of cellular composition in the model used, and electrophysiology measurements in organoids.

The manuscript presents an interesting model of telencephalic in vitro development based on single neural rosettes (SNR) isolation, which allows the generation of organoids with a single lumen bearing predictable organization and cytoarchitecture. The model is then applied to study the alterations promoted by SHANK3 mutation. The authors report excitatory synaptic deficits associated with impaired expression of clustered protocadherins. I have no additional comments on the new experiments, however there remains a single major issue, in my opinion, that makes the comparative analysis of mutant and controls difficult to interpret. The issue that control and mutant lines are not isogenic, and thus it is difficult to establish what differences emerge from the fact that the two preparations are completely different in genetic background versus what is caused by the mutation. The authors derived an iPSC line from a patient with Phelan-McDermid syndrome carrying a partial SHANK3 deletion (Figure 6 and Figure S20)) and then compared it to compare organoids from an unrelated unrelated iPSC line, GM07492. Correction of the mutation in the patient line is the standard these days. Genome editing techniques could have been applied to correct SHANK3 mutation and generate the correspondent isogenic control cell line for proper control. This would have also clarified whether the molecular phenotypes found in the the patient cell line are solely driven by SHANK3 or by other genes present on chromosome 22q13.3 deletion (Phelan-McDermid syndrome). Some recognition of this major limitation must be included as the data cannot fully be interpreted because of these limitations.

Reviewer #3 (Remarks to the Author):

Review of Wang et al. resubmitted to Nature Communications

In their resubmitted manuscript, Wang and colleagues present what they propose is a new method for generating cerebral organoids that contain cells of dorsal telencephalon and LGE lineage. They

perform dual SMAD inhibition of iPS cells in 2D culture, a strategy conventionally used to derive cells of dorsal telencephalon lineage, and then perform manual picking of single neural rosettes, which they culture in 3D. They perform an in-depth characterization of cell types present in their organoids at 1 month and 5 months of development. They confirm that their organoids contain cell types previously described but also highlight that their organoids reproducibly contain cells resembling progenitors of the lateral ganglionic eminence at 1 month of differentiation and a small fraction of striatal medium spiny neurons (~3%) at 5 months of differentiation. They confirm that their organoids develop to become functionally mature using patch clamp electrophysiology, MEA, and immunostaining for synaptic markers. Finally, they apply their system to study Phelad-McDermid Syndrome comparing organoids derived from SHANK3^{+/-} and control iPS lines.

The manuscript is well-written with polished figures, and provides a strong characterization of the cell type composition of their organoids as well as their functional characteristics. However, I do not think that the authors sufficiently addressed my previous comments, and I remain unenthusiastic about the manuscript due to the lack of novelty and benchmarking against existing protocols. I list my comments below which include potential directions the authors could pursue to strengthen their story.

Major comments:

I would hesitate to describe SNOs as a novel method given that a very similar method for generating organoids via single rosette picking has already been described in 2017 (PMID: 27534267). Furthermore, other than the single rosette picking, the authors' method does not differ considerably from current protocols that utilize dual smad inhibition. Cell type composition of organoids derived using dual smad inhibition has already been described previously (for example, PMID: 30573846, PMID: 31168097, PMID: 31996853, to list a few), as well as the emergent functional properties of organoids (for example, PMID: 31474560). Proper laminar localization of neuronal and glial cell types has also been described previously (most recently, PMID: 32142682). The authors' discovery that their protocol yields a small population of LGE-derived medium spiny neurons is novel, but without demonstrating a use case for this population of cells, this discovery would appear to be incremental.

Despite my comment above, I would consider it a worthwhile endeavor to directly benchmark organoids derived using this method against organoids containing multiple rosettes, to showcase the proposed advantages of this new system. As I mentioned in my previous review comments, benchmarking SNOs against multi-rosette organoids to demonstrate the limitations of multi-rosette organoids and how SNOs outperform them would make their paper considerably stronger. The authors have already listed in their paper several limitations of the current protocols that they could compare, including "unpredictable organization of germinal zones and cells in organoids, variability [presumably in the size and shape of the lumen and cortical layers], impaired specification of certain types of cortical NPs and neurons and disrupted cortical layering". Also, authors should consider co-embedding their single cell datasets with published datasets they mention, i.e. Bhaduri et al. 2020, to directly show improvements. While I'm inclined to believe that the authors' method has its advantages compared to current state-of-the-art, it's difficult to ascertain how much of an improvement their method provides without a direct comparison.

Another way the authors could make the paper stronger is by showing more reproducibility data within and across batches and more cell lines, as suggested in the previous round of reviewer comments. For example, how variable are the size and shape of rosettes, lumen, and cortical lamina within and across multiple batches/across multiple lines? How reproducible is the observation that the LGE-derived neurons congregate beneath the cortical lamina within and across multiple batches/across multiple lines? The authors argue that the number of lines they use meets the current standard in the field, but given the lack of apparent novelty, testing across more lines would help strengthen the story considerably.

On a related note, it has been my personal experience that while there are some iPS cell lines that are efficient at generating cortical cells without Wnt inhibition, many others are not efficient at all. I would caution the authors against highlighting the point about how the absence of Wnt inhibition during neural specification is a strong advantage of their protocol without testing in more lines to show reproducibility.

The disease component of the manuscript could be strengthened considerably. Currently the disease component of the story appears superficial and not particularly novel, and seems underpowered with only 1 disease line. Otherwise, a short vignette could be appropriate if it highlighted a unique use case for SNOs that would be difficult to demonstrate using multi-rosette organoids.

We very much appreciate the reviewers' comments on our study. The comments have helped us tremendously to improve the paper. Below, we provide our point-by-point responses (in blue) to the reviewers' comments (in black).

REVIEWER COMMENTS

Reviewer #1 (Remarks to the Author):

The authors did a good job responding to my criticisms. The new protocol is very exciting and the data on shank3 adds new biology to the function of this gene. Thus, I support this publication.

Response: We highly appreciate the reviewers' comments on our study. Thank you so much for your time and thoughtful questions.

Reviewer #2 (Remarks to the Author):

Modeling human telencephalic development and autism-associated SHANK3 deficiency using organoids generated from single neural rosettes

Wang and colleagues addressed most of my original concerns. The manuscript was appropriately edited to address the points that we raised. Substantial new analysis and experiments were added including comparisons with human in vivo transcriptomic data, evaluation of reproducibility of cellular composition in the model used, and electrophysiology measurements in organoids.

The manuscript presents an interesting model of telencephalic in vitro development based on single neural rosettes (SNR) isolation, which allows the generation of organoids with a single lumen bearing predictable organization and cytoarchitecture. The model is then applied to study the alterations promoted by SHANK3 mutation. The authors report excitatory synaptic deficits associated with impaired expression of clustered protocadherins. I have no additional comments on the new experiments, however there remains a single major issue, in my opinion, that makes the comparative analysis of mutant and controls difficult to interpret. The issue that control and mutant lines are not isogenic, and thus it is difficult to establish what differences emerge from the fact that the two preparations are completely different in genetic background versus what is caused by the mutation. The authors derived an iPSC line from a patient with Phelan-McDermid syndrome carrying a partial SHANK3 deletion (Figure 6 and Figure S20)) and then compared it to compare organoids from an unrelated iPSC line, GM07492. Correction of the mutation in the patient line is the standard these days. Genome editing techniques could have been applied to correct SHANK3 mutation and generate the correspondent isogenic control cell line for proper control. This would have also clarified whether the molecular phenotypes found in the the patient cell line are solely driven by SHANK3 or by other genes present on chromosome 22q13.3 deletion (Phelan-McDermid syndrome). Some recognition of this major limitation must be included as the data cannot fully be interpreted because of these limitations.

Response: Thank you for your overall positive evaluation of our revised manuscript. We appreciate your comments. To address the remaining concern, in the revised manuscript, we indicate that it is technically impossible to correct the large chromosomal deletion detected in patient iPSC line (page 23). Therefore, we compared gene expression profiles among this line,

an independent control iPSC line (GM07492), a control ESC line (H9), and an isogenic SHANK3-deficient derived from H9 (EYQ2-20). Importantly, all SHANK3-deficient organoids exhibited impaired expression of clustered protocadherins, as compared to control lines (Fig. 6). This result suggests that this deficit is likely caused by the hemizygous *SHANK3* deletion. In the revised manuscript, we acknowledge the reviewer's concern by adding the following verbiage to the manuscript (page 23):

"The detected deficits will need to be confirmed in organoids from more patient and engineered iPSC lines as it was technically challenging to correct the large chromosomal abnormality in patient iPSC line and to generate additional isogenic SHANK3 deficient lines with a complete hemizygous *SHANK3* deletion."

Reviewer #3 (Remarks to the Author):

Review of Wang et al. resubmitted to Nature Communications

In their resubmitted manuscript, Wang and colleagues present what they propose is a new method for generating cerebral organoids that contain cells of dorsal telencephalon and LGE lineage. They perform dual SMAD inhibition of iPS cells in 2D culture, a strategy conventionally used to derive cells of dorsal telencephalon lineage, and then perform manual picking of single neural rosettes, which they culture in 3D. They perform an in-depth characterization of cell types present in their organoids at 1 month and 5 months of development. They confirm that their organoids contain cell types previously described but also highlight that their organoids reproducibly contain cells resembling progenitors of the lateral ganglionic eminence at 1 month of differentiation and a small fraction of striatal medium spiny neurons (~3%) at 5 months of differentiation. They confirm that their organoids develop to become functionally mature using patch clamp electrophysiology, MEA, and immunostaining for synaptic markers. Finally, they apply their system to study Phelan-McDermid Syndrome comparing organoids derived from SHANK3+/- and control iPS lines.

The manuscript is well-written with polished figures, and provides a strong characterization of the cell type composition of their organoids as well as their functional characteristics. However, I do not think that the authors sufficiently addressed my previous comments, and I remain unenthusiastic about the manuscript due to the lack of novelty and benchmarking against existing protocols. I list my comments below which include potential directions the authors could pursue to strengthen their story.

Response: We appreciate the reviewer's comments on our manuscript and the acknowledgment of the strong characterization of cell types and functional properties of cells in organoids.

We would like to emphasize that the novelty of this study is related to the report of SNT-derived telencephalic organoids with both cortical and striatal neurons. We provide an in-depth characterization of cell types present in these organoids at 1 and 5 months of development using single-cell RNA sequencing and patch-clamp electrophysiology. In addition, the manuscript for the first time describes the functional and molecular deficits in human organoids with the hemizygous *SHANK3* deletion, which is the most common genetic abnormality detected in patients with Phelan-McDermid syndrome.

We also would like to emphasize that we have extensively benchmarked SNR-derived organoids against primary cortical and striatal human brain tissue (Figs. 3i, S12, S14, S15, and S16). To address the reviewer's comment regarding the benchmarking against existing protocols, we performed additional analysis and investigated the expression levels of key cell-type-specific markers in previously reported organoids (Bhaduri et al., 2020; Giandomenic et al., 2019; Xiang et al., 2017; Quadrato et al., 2017; and Velasco et al., 2019) as compared to our organoids (Supplementary Fig. 19). Interestingly, we found that our organoids contain substantially more cells that express GSX2, which is a marker of LGE-derived NPs. This result suggests that GSX2-expressing NPs are required for the specification of primary striatal neurons in telencephalic organoids. Unexpectedly, we also observed reduced proportions of FOXG1- and HOPX-expressing cells in some, but not all, previously published organoid datasets (Supplementary Fig. 19).

Major comments:

I would hesitate to describe SNOs as a novel method given that a very similar method for generating organoids via single rosette picking has already been described in 2017 (PMID: 27534267). Furthermore, other than the single rosette picking, the authors' method does not differ considerably from current protocols that utilize dual smad inhibition. Cell type composition of organoids derived using dual smad inhibition has already been described previously (for example, PMID: 30573846, PMID: 31168097, PMID: 31996853, to list a few), as well as the emergent functional properties of organoids (for example, PMID: 31474560). Proper laminar localization of neuronal and glial cell types has also been described previously (most recently, PMID: 32142682). The authors' discovery that their protocol yields a small population of LGE-derived medium spiny neurons is novel, but without demonstrating a use case for this population of cells, this discovery would appear to be incremental.

Response: We would like to emphasize that a key novelty of this study is related to the generation of telencephalic organoids with both cortical and striatal neurons from SNRs. None of the previous studies reported the presence of both cortical and striatal neurons within the same organoid. This is likely related to the lack of GSX2-expressing progenitors, as we now show in Supplementary Fig. 19. We describe the molecular pathways that are associated with an early specification of GSX2-expressing NPs and LGE-derived inhibitory neurons in SNR-derived organoids.

In the manuscript, we acknowledge (page 22): "in the future, it will be important to investigate how these neurons communicate with each other and whether they form predominantly unidirectional cortico-striatal connections, as it was observed using cortico-striatal assembloids"

Despite my comment above, I would consider it a worthwhile endeavor to directly benchmark organoids derived using this method against organoids containing multiple rosettes, to showcase the proposed advantages of this new system. As I mentioned in my previous review comments, benchmarking SNOs against multi-rosette organoids to demonstrate the limitations of multi-rosette organoids and how SNOs outperform them would make their paper considerably stronger. The authors have already listed in their paper several limitations of the current protocols that they could compare, including "unpredictable organization of germinal zones and cells in organoids, variability [presumably in the size and shape of the lumen and cortical layers], impaired specification of certain types of cortical NPs and neurons and disrupted cortical layering". Also, authors should consider co-embedding their single cell datasets with published datasets they mention, i.e. Bhaduri et al. 2020, to directly show improvements. While I'm inclined to believe that the authors' method has its advantages compared to current state-of-the-

art, it's difficult to ascertain how much of an improvement their method provides without a direct comparison.

Response: We agree with the reviewer that it could be informative to directly compare the properties of single- and multi-lumen-containing organoids. However, this comparison is unlikely to change our main conclusions regarding the specification of different telencephalic cells and disease phenotypes in SNR-derived organoids. Therefore, we were hesitant to perform the comparison experiments. We hope the reviewer will agree with us that those experiments are not necessary for supporting the main conclusions of this study and that the efforts required to conduct those experiments could easily constitute a whole different paper.

In the revised manuscript, we acknowledge this comment by stating (page 23): “Additional studies will be needed to understand how the batch-to-batch, line-to-line, and individual-to-individual differences influence the reproducibility of germinal zone organization, size and shape of the lumen and cortical layers, and specification of different telencephalic cell types in organoids produced from single and multiple rosettes.”

We appreciate the reviewer's suggestion to directly compare the gene expression profiles of our organoids with those reported in the previous studies. We specifically investigated the expression profiles of key cell type identity markers, such as FOXP1 (telencephalic), PAX6 (pallial NPs), GSX2 (subpallial NPs), HOPX (oRG), STMN2 (neurons), RBFOX3 (mature neurons), LMO3 (deep-layer cortical excitatory neurons), SATB2 (superficial-layer cortical excitatory neurons), GAD1/2 (inhibitory neurons), AQP4 (astrocytes), S100B (astrocytes and ependymal cells), in several previously published datasets to compare and contrast them with the profile in our organoids (Supplementary Fig. 19). Interestingly, we found that SNR-derived organoids contain increased proportions of GSX2-expressing cells, which may explain the absence of primary striatal neurons in those other studies.

Another way the authors could make the paper stronger is by showing more reproducibility data within and across batches and more cell lines, as suggested in the previous round of reviewer comments. For example, how variable are the size and shape of rosettes, lumen, and cortical lamina within and across multiple batches/across multiple lines? How reproducible is the observation that the LGE-derived neurons congregate beneath the cortical lamina within and across multiple batches/across multiple lines? The authors argue that the number of lines they use meets the current standard in the field, but given the lack of apparent novelty, testing across more lines would help strengthen the story considerably.

Response: We appreciate these suggestions. In the revised manuscript, we added data for two more lines and more batches to further characterize the reproducibility in 1-month-old organoids (Supplementary Fig. 5).

We also made a new figure with the characterization of single rosettes obtained from different lines across multiple differentiation batches (Supplementary Fig. 2).

Finally, we performed additional immunostaining analyses to characterize the distribution of Ctip2- and Ctip2/DARPP32/GABA-expressing cells in organoid sections produced from different stem cell lines across several differentiation batches (Fig. 4g and Supplementary Fig. 17). The results confirm our previous observations regarding the congregation of LGE-derived neurons beneath the cortical lamina in 5-month-old SNR-derived organoids.

On a related note, it has been my personal experience that while there are some iPS cell lines that are efficient at generating cortical cells without Wnt inhibition, many others are not efficient

at all. I would caution the authors against highlighting the point about how the absence of Wnt inhibition during neural specification is a strong advantage of their protocol without testing in more lines to show reproducibility.

Response: Thank you for this note. In the revised manuscript, we indicate that Wnt inhibitors and RA agonists have been used for an efficient specification of cortical and striatal cells in iPSC-derived organoids, respectively (page 4).

The disease component of the manuscript could be strengthened considerably. Currently the disease component of the story appears superficial and not particularly novel, and seems underpowered with only 1 disease line. Otherwise, a short vignette could be appropriate if it highlighted a unique use case for SNOs that would be difficult to demonstrate using multi-rosette organoids.

Response: We appreciate the reviewer's concern, and in the revised manuscript, we describe the following limitations of our study (page 23):

"The detected deficits will need to be confirmed in organoids from more patient and engineered iPSC lines as it was technically challenging to correct the large chromosomal abnormality in patient iPSC line and to generate additional isogenic SHANK3 deficient lines with a complete hemizygous *SHANK3* deletion."

REVIEWERS' COMMENTS

Reviewer #3 (Remarks to the Author):

Wang and colleagues have sufficiently addressed the concerns I had outlined in my review of their manuscript. I am satisfied with the additional data they provide that support the reproducibility of their claims, including their comparisons with previous scRNA-seq datasets that suggest an enrichment of LGE-derived celltypes in their single rosette-derived organoids, as well as the changes they made to the text in acknowledgement of the feedback provided. I am also satisfied with their clarification with regards to the novelty of their work that I had not previously appreciated, i.e. that their study is the first to characterize a SHANK3 hemizygous knockout organoid model. In summary, I would support the publication of this manuscript.

Response to Referees:

REVIEWERS' COMMENTS

Reviewer #3 (Remarks to the Author):

Wang and colleagues have sufficiently addressed the concerns I had outlined in my review of their manuscript. I am satisfied with the additional data they provide that support the reproducibility of their claims, including their comparisons with previous scRNA-seq datasets that suggest an enrichment of LGE-derived celltypes in their single rosette-derived organoids, as well as the changes they made to the text in acknowledgement of the feedback provided. I am also satisfied with their clarification with regards to the novelty of their work that I had not previously appreciated, i.e. that their study is the first to characterize a SHANK3 hemizygous knockout organoid model. In summary, I would support the publication of this manuscript.

Response: We highly appreciate the reviewer's comments on our paper. They have helped us tremendously to improve the paper. Thank you for your time and constructive critique.